# Interfacial chemistry-driven reaction dynamics and resultant microstructural evolution in lithium-based all-solid-state batteries

Achieving a comprehensive understanding of battery systems necessitates multi-length scale analysis, from the atomic- to macro-scale, to grasp the complex interplay of phenomena influencing performance. However, studies to understand these phenomena in all-solid-state batteries (ASSBs) poses significant challenges due to the complex microstructural evolution involved, including the pore formation and contact loss resulting from cathode material breathing, chemical degradation at interfaces, and their interplay. Herein, we investigate the impact of chemical degradation on the reaction behavior and microstructural evolution of Ni-rich cathode particle ($LiNi_{0.6}Co_{0.2}Mn_{0.2}O_2$) within composite cathodes of sulfide-based ASSBs, using a well-defined model system incorporating Li-In alloy anodes and a non-decomposable coating layer that solely alters the interfacial chemical reactivity. By using lithium difluorophosphate (LiDFP) to suppress chemical degradation, we observed that this suppression enhances the reaction uniformity among particles and homogenizes mechanical degradation, albeit increasing pore formation and tortuosity. In addition, unbridled chemical degradation induces significant reaction heterogeneity and non-uniform mechanical degradation, with fewer pores and lower tortuosity. These findings complement the understanding of mechanical degradation, which is traditionally described using the metrics of contact loss and tortuosity, and underscore the critical role of coating layers in promoting lithium conduction by maintaining contact with the cathode surface.

Advancing battery systems demands a comprehensive understanding across multiple length scales, from atomic to meter level[1]. Crucial to this is grasping the interplay among various phenomena at various scales, from laboratory to industrial, and their collective impact on battery systems and performance - a prerequisite for rapid technological development[2,3]. This necessitates in-depth knowledge of active-material particles and electrode behavior at the micrometer scale[4], particularly focusing on material thermodynamics and kinetics under diverse operating conditions[5,6]. Recent years have seen significant strides in elucidating these aspects in lithium-ion batteries (LIBs). Groundbreaking research has unveiled lithium heterogeneities at both intra- and inter-particle levels during high-rate charge/discharge conditions[7–9]. These findings have deepened our understanding of mass-transfer kinetics in cathode materials and charge-transfer kinetics within the LIB system[10]. Such insights are pivotal in developing charge/discharge protocols that promote homogeneous reactions within electrodes, thereby minimizing cathode particle degradation and extending cycle life[11]. This progress underscores the critical

✉ e-mail: jhong@postech.ac.kr; dkim@unist.ac.kr; naecard@snu.ac.kr

importance of a multi-scale understanding, from particle to electrode level, in optimizing battery systems. By bridging these scales, researchers can develop more efficient, durable, and high-performance battery technologies to meet growing energy storage demands.

Next-generation batteries, particularly all-solid-state batteries (ASSBs), demand a similar multi-scale understanding[12]. However, progress in this field has been hindered by its significantly higher complexity compared to LIBs[13,14]. In particular, the reaction behavior of Ni-rich cathode particles during ASSB operation inevitably involves changes in the composite microstructure with the solid electrolyte, such as pore formation, loss of contact, and variations in tortuosity, due to the breathing of cathode materials[15–18]. Unlike LIBs, where contact between active material and electrolyte remains constant, ASSBs experience dynamic contact changes[19]. This dynamic nature could result in a change of the effective current density experienced by particles over time, continuously exposing individual particles to various operating environments. As a result, the reaction behavior of cathode particles and the ensuing microstructural changes form a highly complex, and interdependent system. Further complicating matter is chemical degradation at the cathode–solid-electrolyte interface. This degradation not only slows charge-transfer kinetics but also influences mechanical-degradation patterns[20]. The interplay between these chemical and mechanical degradation processes creates a dual challenge, making it exceptionally difficult to simultaneously comprehend the reaction behavior of cathode materials and microstructure changes within ASSBs.

To achieve a comprehensive multi-scale understanding of ASSBs, it is essential to first elucidate how chemical degradation affects particle reaction behaviors and microstructural changes within electrodes. In this study, we aimed to investigate the impact of chemical degradation at the cathode|solid-electrolyte interface on the reaction behaviors and microstructural changes of Ni-rich cathode particle ($LiNi_{0.6}Co_{0.2}Mn_{0.2}O_2$, NCM) in sulfide-based ASSBs. To elucidate the intertwined chemo-mechanical behavior of ASSBs, we utilized a model system comprising single-crystalline cathodes and sulfide-based solid electrolytes ($Li_6PS_5Cl$), which offer significant advantages. While polycrystalline cathodes are widely used for their high tap density and cost advantages, they suffer from intergranular cracking and internal pore formation during cycling and pressing, leading to mechanical degradation and electrochemical isolation of cathode particles—both of which adversely affect ASSB performance[21,22]. In contrast, single-crystal cathodes, free from intergranular cracking, offer a more crack-resistant architecture that mitigates these issues and enables clearer observation of chemically driven interfacial effects[23]. Furthermore, oxide-based electrolytes must undergo high-temperature annealing, which often densifies the interface and produces undesirable secondary phases at the cathode[24,25]. By contrast, sulfide solid electrolytes (SEs) can be processed at low temperatures, enabling a conformal and intimate contact with the cathode particles. Their relatively compliant mechanical nature minimizes artificial fracture or densification effects, allowing more accurate observation of chemically induced interfacial and morphological changes[26]. We employed lithium difluorophosphate (LiDFP, $LiPO_2F_2$) as a model coating material to suppress chemical degradation[27]. Unlike conventional oxide-based coatings such as $LiNbO_3$ or $Li_4Ti_5O_{12}$, which often undergo oxidative decomposition or structural changes under high-voltage and high-pressure conditions[20,28,29], LiDFP forms a chemically and electrochemically stable interfacial layer that remains intact even during the cycling at elevated voltages[30]. This stability enables more reliable analysis of reaction uniformity and microstructural changes in composite electrodes with and without the coating. Our methodology combined in-situ X-ray diffraction (XRD) and transmission X-ray microscopy (TXM) to compare reaction homogeneity among particles. We analyzed the internal structure of the cathode composite electrode using 3D digital

twins acquired from focused-ion-beam scanning electron microscopy (FIB-SEM) imaging. Furthermore, we conducted quantitative analysis of porosity, pore size, and distribution around hundreds of cathode particles to ensure statistical significance.

Our findings reveal that suppressing chemical degradation through an intact interfacial layer formed by coating materials significantly enhanced reaction uniformity among particles, leading to homogenized mechanical degradation. However, this resulted in increased pore formation and tortuosity in the composite electrode structure compared to uncoated electrodes. Conversely, unsuppressed chemical degradation, combined with the intrinsically heterogeneous interfacial point-contact, led to increased reaction heterogeneity among particles, causing non-uniform mechanical degradation but with fewer pores and lower tortuosity. These results underscore that microstructure change conventionally represented by contact loss or tortuosity needs to be complemented to more clearly explain the mechanical degradation of ASSBs. Our study also underscores the potential of coating layer to maintain active surface area by providing a large-dimension lithium-conduction pathway to the cathode, contrasting with the geometric point contact between cathode and solid electrolyte. This work thus offers fresh insights into chemo-mechanical degradation in ASSBs and strategies for its mitigation, advancing our understanding of multi-scale interactions in next-generation battery systems. This study provides a more nuanced perspective on the complex interplay between chemical and mechanical processes in advanced energy storage devices, guiding the development of high-performance ASSBs.

## Results and discussions
### Model system for selectively suppression of chemical degradation and its impact on electrochemical performance

To investigate the impact of chemical degradation at the cathode|solid-electrolyte interface on cathode particle reaction behavior and microstructural changes, we prepared two cathode types: one without surface treatment and another with an electronically insulating coating to mitigate oxidation decomposition of SEs[31]. Commonly used inorganic oxide coating materials like $LiNbO_3$ and $LiZr_2O_3$ have limitations in withstanding mechanical stress from cathode volume changes due to their high Young's modulus[32,33], as well as the possibility of oxidation reactions under high-voltage conditions[15], which may hinder a sophisticated understanding of chemo-mechanical evolution at the interface between cathode and SE. Thus, we selected the Li-salt compound LiDFP due to its robustness in high-voltage environments and relatively lower Young's modulus[34,35]. We employed the mechano-fusion method, which utilizes shear force friction between cathode and coating materials to create a uniform interfacial layer without morphological or compositional changes (Supplementary Fig. 1)[20]. TEM analysis confirmed the presence of a homogeneous and thin LiDFP-based film (approximately 10 nm thick) on the cathode surface (Supplementary Fig. 2a, b). Post mechano-fusion process, we used time-of-flight secondary ion mass spectrometry (TOF-SIMS) to analyze the coating composition. Consistent peaks of $PO_2^-$ (~62.97 u) and $PO_2F_2^-$ (~100.96 u) anions, corresponding to LiDFP signals, were observed. This indicated successful application of the coating materials without compositional alterations (Supplementary Fig. 2c, d). The LiDFP effectively alters only the electronic conductivity within the cathode composite, resulting in a significant decrease by an order of magnitude for LiDFP $LiNi_{0.6}Co_{0.2}Mn_{0.2}O_2$ (NCM) ($2.74 \times 10^{-9}$ S/cm vs. $6.90 \times 10^{-8}$ S/cm for bare NCM) (Fig. 1a and Supplementary Table 1), while maintaining a comparable ionic conductivity ($9.37 \times 10^{-5}$ S cm$^{-1}$ vs. $9.19 \times 10^{-5}$ S cm$^{-1}$ for bare NCM), as shown in Fig. 1b and Supplementary Table 2. It implies that electronically insulating LiDFP well covered the cathode surface and microstructure regarding ionic percolating network is maintained regardless of incorporation of coating layer. This is further supported by the morphological analysis of the

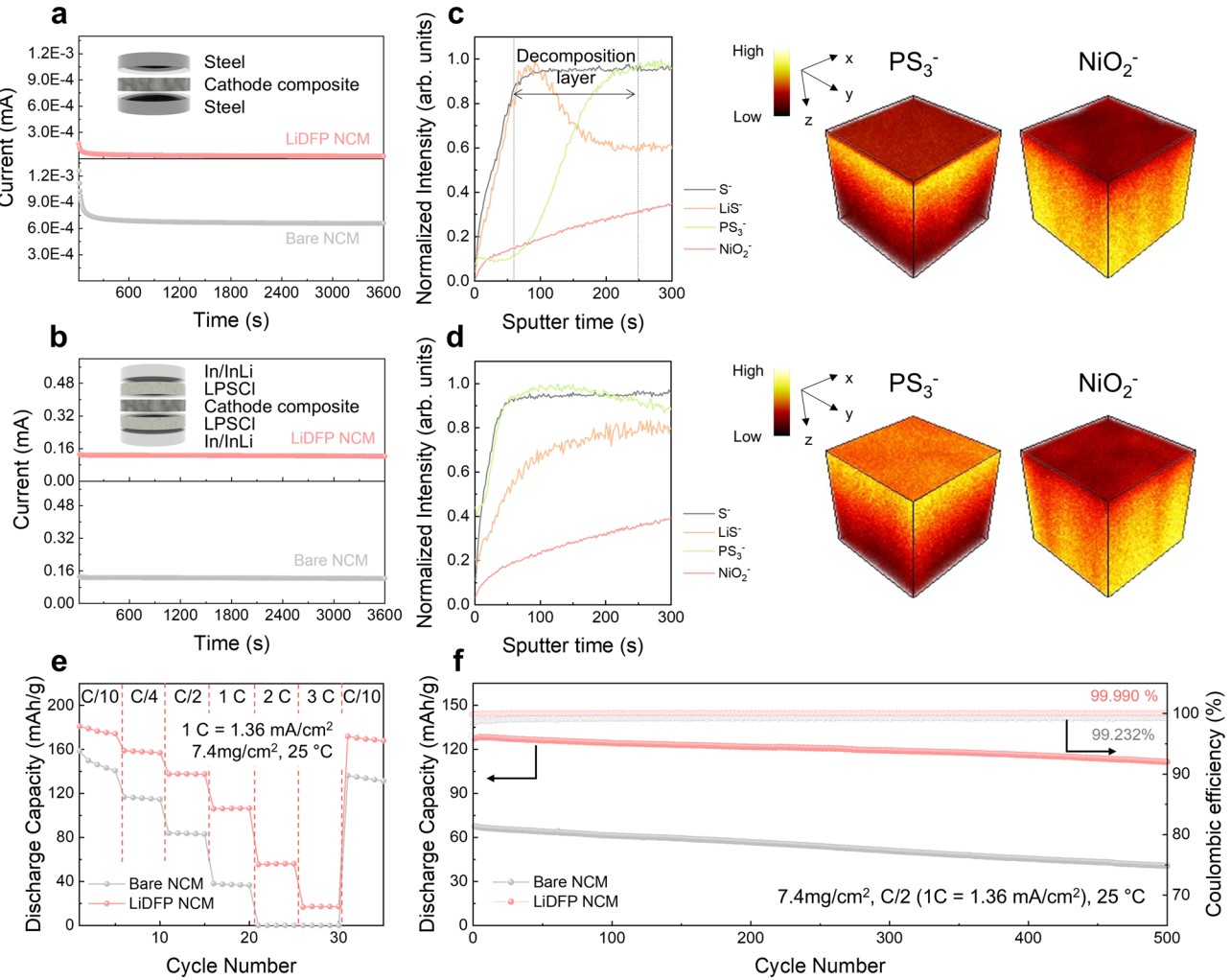

**Fig. 1 | Chemical degradation mitigation through surface modification and corresponding electrochemical properties. a** Current–time curves of the cathode-SE composite (65:35 wt%) under a polarization of 200 mV in a steel|NCM composite|steel ion-blocking cell, and **b** current–time curves of the cathode-SE composite (65:35 wt%) under a polarization of 50 mV in an electron-blocking cell, composed of In/InLi|LPSCl|NCM composite|LPSCl||In/InLi, for both bare NCM and LiDFP NCM. TOF-SIMS 3D depth spectra and 3D images of **c** bare NCM and **d** LiDFP NCM for S⁻, LiS⁻, PS₃⁻, and NiO₂⁻ anion after the first cycling. Comparison of **e** rate capability of bare NCM and LiDFP NCM at varying C-rates: C/10, C/4, C/2, 1 C, 2 C, 3 C, and back to C/10 and subsequent **f** long-term cycle performance at C/2 between bare NCM and LiDFP NCM at 25 °C under a stack pressure of 120 MPa with 7.4 mg/cm² active material mass loading.

composite pellet, which shows no significant differences in pore formation induced by interfacial modification between the bare NCM and LiDFP NCM (Supplementary Fig. 3). Moreover, the LiDFP is well maintained even under high-voltage conditions in contact with sulfide SEs, which is demonstrated by negligible oxidation (Supplementary Fig. 4). Additionally, after the initial cycling, the LiDFP on the surface of the cathode particles remains intact, as shown in Supplementary Fig. 5. These findings confirm that interfacial modification with LiDFP can be a model system that selectively controls the chemical reactivity at the interface, without microstructural changes when forming the composite cathode with SEs.

The effect of chemical degradation at the interface on electrochemical properties was first compared depending on surface modification. As depicted in Supplementary Fig. 5, the LiDFP coating interfacial protective layer significantly enhances reversible capacity, surpassing that of bare NCM by over 15 mAh/g. Moreover, it exhibits a higher coulombic efficiency of 80.5% compared to 74.8% for bare NCM. To further validate LiDFP chemical stability in high-voltage environments, the cells were charged to a high voltage of 3.88 V (vs. In/InLi; corresponding to 4.5 V vs. Li⁺/Li), and impedance changes were measured during the subsequent rest period (Supplementary

Figs. 6 and 7). These precise impedance measurements elucidated the chemical degradation kinetics at the interfacial layer between the cathode active materials (CAM) and the argyrodite (Li₆PS₅Cl, LPSCl) electrolyte. The distribution of relaxation times (DRT) method was employed to distinguish resistances across various frequencies, extracting the interfacial resistance at the CAM|SE interface from the middle-frequency range (Supplementary Fig. 8)[20,36,37] Assuming ion diffusion governs charge transport across the interfacial layer, the CAM|SE interfacial resistance ($R_{CEI}$) increases linearly with the square root of time ($t^{0.5}$) under open-circuit-voltage conditions. This behavior is described by diffusion-controlled model Eq. (1), with specific parameters listed in Supplementary Table 3[38].

$$R_{CEI} = \frac{1}{S\overline{\sigma_{CEI}}} \sqrt{\frac{V_m}{xF^2} \cdot \frac{\overline{\sigma_{Li^+} \cdot \sigma_{e^-}}}{\overline{\sigma_{Li^+} + \sigma_{e^-}}} \Delta\mu_{Li}} \cdot \sqrt{t} = \frac{1}{S\overline{\sigma_{CEI}}} \cdot k\sqrt{t} = k'\sqrt{t} \quad (1)$$

The behavior of $R_{CEI}$ differs significantly between bare NCM and LiDFP-coated NCM (Supplementary Fig. 7c). Bare NCM exhibits a quadratic increase in $R_{CEI}$ with the square root of time (11.86 Ω h⁻¹), indicating accelerating degradation. In contrast, LiDFP NCM shows a linear increase in $R_{CEI}$ (50.57 Ω h⁻⁰·⁵), aligning with the diffusion-

controlled model. Moreover, LiDFP NCM demonstrates a lower initial resistance of 165.7 Ω compared to 209.5 Ω for bare NCM. These results strongly suggest that LiDFP protective layer forms a more stable interface with the sulfide SE than bare NCM, effectively mitigating chemical degradation.

TOF-SIMS analysis conducted after initial formation cycles consistently verifies the observed trends (Fig. 1c, d). For bare NCM, the moiety associated with $PS_3^-$ ions, indicative of the argyrodite SE anion framework, only reach saturation after a sputter time of 240 s. 3D imagery reveals the diminution of the $PS_3^-$ fraction at the surface before encountering the NCM cathode (denoted as $NiO_2^-$) (Fig. 1c). The disparity between $S^-$ and $PS_3^-$ ion saturation points is attributed to the accumulation of sulfur-based byproducts such as $LiS^-$, resulting from sulfide SE oxidation. This observation signifies substantial chemical degradation product formation on the bare NCM surface, highlighting severe oxidative decomposition of the SE. Notably, this precipitous increase in the interfacial byproduct layer could be particularly pronounced at point contacts between the cathodes and the SE, manifesting as localized byproduct formations. In stark contrast, LiDFP NCM exhibits simultaneous saturation of $S^-$ and $PS_3^-$ ions, which originated from LPSCl, from the outset, without a noticeable increase in $LiS^-$ byproduct. This simultaneous saturation, depicted in Fig. 1d, indicates significant mitigation of chemical degradation at the CAM|SE interface by the protective layer, ensuring a more stable interfacial state, except for the LiDFP layer at the surface. These findings imply that the coated NCM possesses an evenly constituted interfacial layer composed of LiDFP, whereas the bare NCM manifests severe byproduct layer evolution at the SE interface. This further corroborates the effectiveness of the LiDFP coating in suppressing chemical degradation and maintaining interface stability.

Electrochemical performance analysis reveals significant differences between bare NCM and LiDFP NCM. Under mild stack-pressure conditions (13 MPa), bare NCM exhibits a higher overpotential of 0.2 V, whereas LiDFP NCM shows a markedly lower overpotential of 0.14 V during cycling (Supplementary Fig. 9). This difference stems from continuous interfacial side reactions at the cathode|solid-electrolyte interface in bare NCM, leading to increased overpotential and rapid capacity decline. Long-term cycling performance at C/4 under mild stack-pressure conditions further illustrates this trend. LiDFP NCM demonstrates an impressive 84% capacity retention after 200 cycles, whereas bare NCM retains only 26.9% of its capacity due to continuous interfacial degradation (Supplementary Fig. 10). In a high-pressure environment (120 MPa), rate capability tests revealed that the superior performance of LiDFP NCM. It not only exhibited higher capacity across all rates but also showed larger capacity increases at higher rates compared to bare NCM (Fig. 1e and Supplementary Fig. 11a, c). The effectiveness of the LiDFP coating is further evidenced in long-term cycling at C/2, where the LiDFP-coated cathode displays a high initial capacity of 127.3 mAh/g compared to 67.6 mAh/g for bare NCM, and it maintains 87.5% of capacity retention after 500 cycles, compared to 59.9% for bare NCM (Fig. 1f and Supplementary Fig. 11b, d). These observations collectively demonstrate that LiDFP functions as an effective protective layer, suppressing chemical degradation while enhancing lithium-ion mobility at the CAM-SE interface. This dual functionality significantly improves the overall electrochemical performance and longevity of the battery.

**Chemical degradation effect on reaction behavior of cathode: in-situ XRD and TXM analysis of lithium heterogeneity**
We investigated the reaction heterogeneity in cathodes with and without surface modification using in-situ XRD and TXM[39,40]. To gain a more detailed understanding of the cathode behavior, we specifically analyzed the (003) reflection of the layered oxide structure within the 2θ range of 18–19°, using in-situ XRD in reflection mode (Supplementary Fig. 12). This analysis, depicted in Fig. 2, elucidated the impact

of chemical degradation on cathode particle reaction behavior in ASSBs. At a slow current density (C/5), bare NCM and LiDFP NCM exhibited nearly identical voltage profiles and charge capacities (202 mAh/g), with a slight difference in coulombic efficiency due to chemical degradation (79.9% for LiDFP NCM vs. 75.1% for bare NCM) (Fig. 2a). However, during galvanostatic evaluation within the voltage range of 1.1.88–3.88 V (vs. In/InLi; corresponding to 2.5–4.5 V (vs. Li⁺/Li)), bare NCM exhibited noticeable (003) peak separation mid-charge, indicating lithium heterogeneity (Fig. 2b and Supplementary Fig. 13a). This phenomenon resembles fictitious phase separation of the layered cathode in LIBs, due to the autocatalytic effect driven by SOC deviation and charge-transfer kinetics[9]. Remarkably, this phase separation, typically observed in liquid-electrolyte systems at high current densities (4C), occurred in ASSBs at low reaction rates (C/5) without surface protection. Conversely, LiDFP NCM displayed typical solid-solution behavior without significant diffraction peak separation (Fig. 2c and Supplementary Fig. 13b). To quantify lithium heterogeneity, we deconvoluted the (003) peak (Supplementary Fig. 14) and subsequently defined the degree of heterogeneity (DOH) at each SOC. We designated the position of each deconvoluted peak at its highest intensity as $x_n$, its area as $w_n$, and the total peak area at a specific SOC as $A$. Given that the area of each resolved peak is proportional to the population of cathode particles at each SOC respectively, we employed statistics based on the weighted average and deviation from the mean peak position ($x_m$) according to SOC, as shown in Eqs. (2) and (3), with the final DOH calculated based on the weighted variance according to Eq. (4). To enhance the visualization of the DOH quantification, the weighted variance values were scaled up by a factor of 10,000.

$$x_m(\text{weighted mean, mean peak position}) = \frac{\sum x_n w_n}{A} \quad (2)$$

$$\sigma_i(\text{weighted deviation}) = \sum w_n \times (x_m - x_n)^2 \quad (3)$$

$$\sigma_i^2(\text{weighted variance}) = \frac{\sum w_n \times (x_m - x_n)^2}{A} \quad (4)$$

To validate the statistical quantification, we monitored the mean peak position change as a function of SOC. As shown in Supplementary Fig. 15a, both bare NCM and LiDFP NCM display similar SOC variations during charge and discharge. Moreover, these changes in ASSBs align with results from liquid-electrolyte LIBs, as evidenced by synchrotron XRD analysis (Supplementary Fig. 15b). Therefore, this consistency confirms that our in-situ XRD and quantitative analysis reliably represent average electrode processes. As shown in Fig. 2b, c, there are striking differences in DOH between bare NCM and LiDFP NCM. The bare NCM showed elevated DOH values exceeding 50 at SOC 40, SOC 80, and end of discharge after the first cycle (SOC 24.9 for bare NCM SOC 20.1 for LiDFP NCM). In contrast, LiDFP NCM maintained DOH values below 40 across all SOC ranges, indicating a more uniform lithium distribution throughout the cathode material (Fig. 2d). We attribute the significant heterogeneity in bare NCM during cycling to chemical degradation. Specifically, byproduct layer formation from solid electrolyte decomposition at the cathode|solid-electrolyte interface increases interfacial resistance and impedes charge-transfer kinetics. In addition, chemical-reaction-induced mechanical degradation[41], resulting from volume reduction with consecutive reactions (as described by reaction Eqs. (5)–(7)) exacerbates charge-transfer kinetics deviation among cathode particles[42–44]. Random point contacts between CAM and SE further

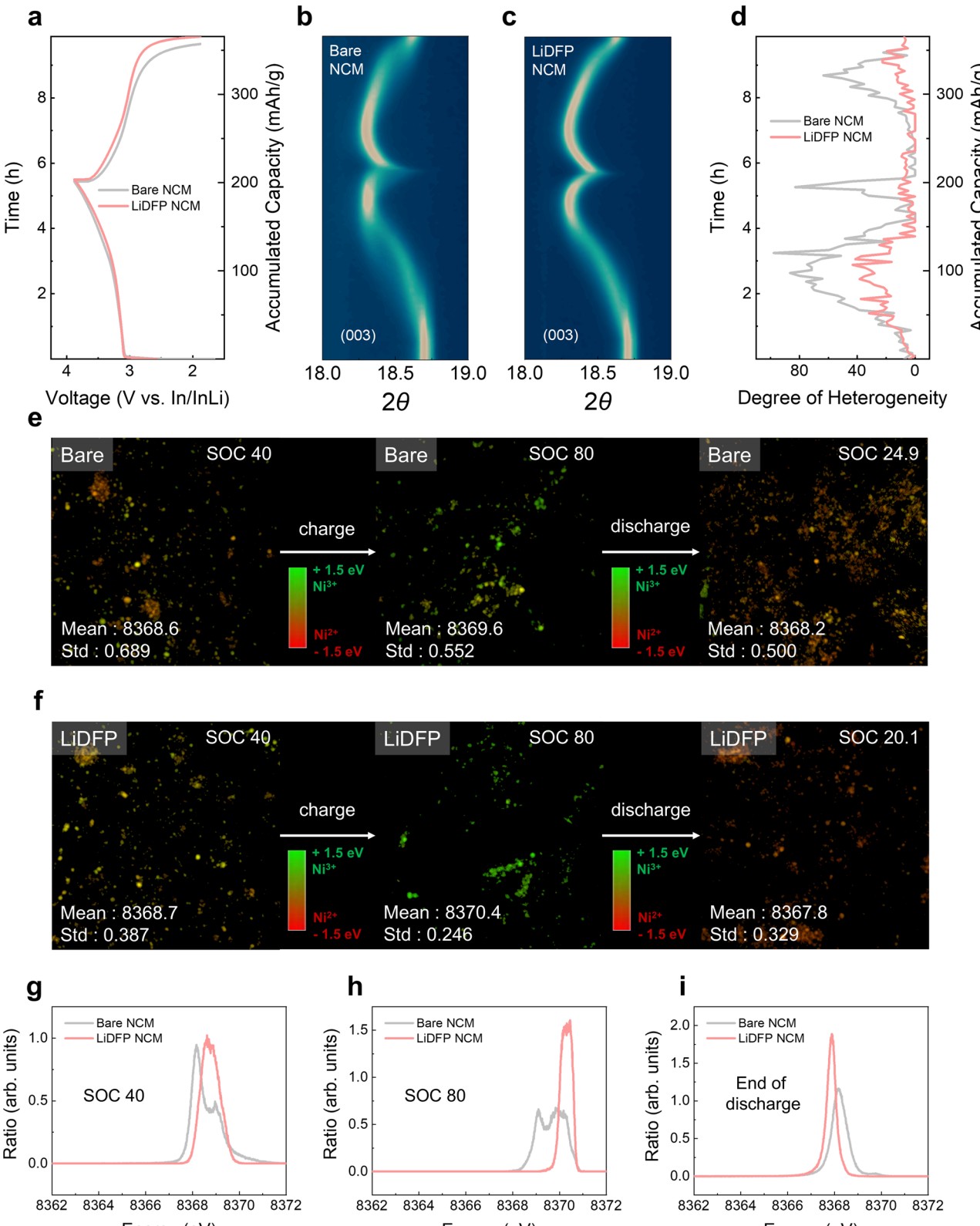

**Fig. 2 | Cathode reaction heterogeneity as influenced by chemical degradation. a** Voltage profile of In/InLi|LPSCl|NCM composite with bare NCM and LiDFP NCM within the voltage range of 1.88–3.88 V (vs. In/InLi) under a current density of 0.48 mA/cm² at 25 °C with a stack pressure of 13 MPa, along with a corresponding contour plot depicting the diffraction peak evolution of the (003) planes for **b** bare NCM and **c** LiDFP NCM during in-situ XRD measurement. **d** Corresponding Degree of Heterogeneity (DOH) parameter acquired from in-situ XRD measurement. TXM-XANES mapping at low magnification with multiple cathode particles at SOC 40, SOC 80, and end of discharge after the first cycle for **e** bare NCM and **f** LiDFP NCM with mean maximum energy value and standard deviation. Corresponding normalized energy value histogram of pixel from TXM image in **g** SOC 40, **h** SOC 80, and **i** end of discharge after the first cycle.

contribute to heterogeneous charge-transfer kinetics.

$$7Li_6PS_5Cl \xrightarrow{-14 Li^+} 7Li_3PS_4 + 7LiCl + 7S \qquad (5)$$

$$7Li_3PS_4 + 24LiNiO_2 \rightarrow 8Ni_3S_2 + 7Li_3PO_4 + 5Li_2SO_4 + 7Li_2S \qquad (6)$$

$$7Li_6PS_5Cl + 24LiNiO_2 \rightarrow 7LiCl + 7S + 8Ni_3S_2 + 7Li_3PO_4 + 5Li_2SO_4$$
$$+ 7Li_2S(\Delta V = -18.45\%) \qquad (7)$$

As shown in Supplementary Fig. 16, a more pronounced (003) peak separation was observed in the aged bare NCM cell after 72 h of assembly without operation, compared to the fresh cell in Fig. 2b. This observation strongly supports the hypothesis that chemical degradation induces lithium heterogeneity in ASSBs. This low oxidative stability of sulfide SE leads to increased interfacial resistance merely through contact with the CAM, even in the absence of current flow[29,41,45]. This finding has two significant implications. First, lithium heterogeneity in ASSBs is primarily driven by reaction-limited conditions rather than bulk diffusion-limited conditions. Second, interparticle lithium heterogeneity can occur and be observed, highlighting the complex nature of charge distribution in these systems. These insights underscore the critical role of interfacial stability in ASSB performance and emphasize the need for effective protective measures, such as the LiDFP coating, to mitigate chemical degradation and maintain reaction homogeneity.

The pronounced increase in DOH due to chemical degradation, initially observed through in-situ XRD, was further corroborated by examining lithium heterogeneity among cathode particles. We employed synchrotron-based TXM to meticulously assess the distribution of Ni oxidation state via Ni K-edge X-ray absorption near-edge structure (XANES) spectra for each 133 nm pixel, using in-house beamline software (Fig. 2e, f)[46,47]. We first plotted particle-averaged oxidation states of Ni ions ($Ni^{2+}/Ni^{3+}$) as a function of SOC in a liquid-electrolyte NCM cathode system (Supplementary Fig. 17). The mean energy from TXM images represents the averaged peak position energy in XANES, which is proportional to the cathode particle SOC, while the standard deviation of this peak position energy reflects lithium heterogeneity distribution among particles. High DOH scores for bare NCM are corroborated by similar standard deviation trends in TXM analysis (Fig. 2d, e), confirming significant heterogeneity. LiDFP NCM consistently exhibits lower values in both DOH scores and TXM standard deviation across different SOC levels compared to bare NCM, indicating more homogeneous reactions among cathode particles (Fig. 2d, f). Bare NCM showed substantial dispersion in Ni oxidation state distribution, with a bifurcated normal distribution within SOC 40 and SOC 80, indicating significant particle-level heterogeneity (Fig. 2g, h). At the end of discharge after the first cycle, bare NCM follows a general normal distribution, but with considerable width due to charge heterogeneity and elevated average SOC, reflecting an increased irreversible capacity (Fig. 2i). In contrast, LiDFP NCM maintains a uniform normal distribution of Ni oxidation states across all SOC levels, even at SOC 40 where a relatively high DOH is observed. These findings strongly suggest that mitigating interfacial degradation effectively enhances reaction homogeneity of cathode particles in ASSBs, contributing significantly to improved cycle stability. The LiDFP coating proves instrumental in maintaining uniform reaction conditions, thereby potentially extending battery life and performance.

### Advanced 3D digital twin analysis for microstructure evolution in ASSBs

The volume change of the cathode during battery operation significantly impacts mechanical degradation at the CAM−SE interface and microstructure evolution in the cathode-solid-electrolyte composite. To comprehensively understand these microstructural changes, it is crucial to consider the reaction behavior of cathode materials affected by chemical degradation. Various imaging analytical techniques, including modeling, cross-sectional scanning electron microscopy (SEM), and micro-computed tomography (micro-CT), have been employed to investigate mechanical degradation and microstructural changes in ASSBs[18,48–50]. However, these approaches have significant limitations. They primarily focus on quantifying and tracking overall pore formation and evolution within the structure, without specifically identifying pores in direct contact with the cathode. Furthermore, the impact of chemical degradation on mechanical degradation is often overlooked. Our previous efforts to map pore distributions using cross-sectional SEM and micro-CT imaging were similarly constrained, providing only a holistic view of pore generation without the ability to discern pores explicitly caused by cathode effects[20]. To overcome these limitations and to account for the heterogeneity of volume changes among cathode particles, a more comprehensive approach is necessary. This approach requires the identification and labeling of hundreds or thousands of individual cathode particles inside the composite electrode. Such a detailed analysis would enable a more detailed understanding of the interplay between chemical degradation, mechanical degradation, and microstructural changes, thus providing deeper insights into the complex dynamics of ASSBs.

To comprehensively analyze and understand the mechanical degradation directly precipitated by the cathode, we employed advanced 3D analytical techniques combining focused−ion−beam−SEM (FIB−SEM) analysis with digital twin technology. This approach aims to provide unprecedented insights into the intricate interplay between the CAM and SE. For the experimental procedure, ASSB pellets, post−formation cycle, were positioned at a 36 ° tilt to ensure perpendicular ion beam incidence on the cathode composite. Initially, the ion beam created islands around the target area, followed by SEM imaging to capture the 2D morphology of the cathode composite, highlighted with a red-dotted rectangle (Fig. 3a). Successive thin slices (97.52 nm) were removed along the z-axis using the ion beam, with SEM images captured after each removal. The resulting image series were then aligned, cropped, and compiled into a 2D image stack (Fig. 3a). A Fourier transform filter was applied to ameliorate the curtaining effect induced during the FIB processing[51]. A U-net algorithm was employed to precisely distinguish and demarcate the gray values of the CAM, SE, carbon, and pores (Fig. 3b and Supplementary Figs. 18–20)[52,53]. Consecutive images were stacked to reconstruct the digital twinned 3D model of the cathode composites for both bare NCM and LiDFP NCM. The final 3D reconstructions were categorized into four distinct components, each with dimensions of 28.28, 39.08, and 55.82 μm along the x, y, and z axes, respectively (Fig. 3c, d). This detailed 3D model enables a comprehensive analysis of the microstructural changes and mechanical degradation in the ASSB system.

While the reconstructed 3D images provide insights into traditional metrics such as porosity and ion/electron tortuosity, they primarily address macro-scale mechanical degradation of the entire electrode. To capture detailed micro-scale behavior at the particle level, we developed a more refined analytical approach. Our analysis focused on key pores impacting the mechanical evolution of ASSBs within the cathode composite, including both inherent pores in the solid electrolyte and cathode particles, as well as voids generated at the CAM−SE interface during galvanostatic cycling. To accurately delineate pores adjacent to NCM particles, we employed a precision-engineered filter window centered on each NCM contours coordinate, enabling comprehensive assessment of neighboring pixels next to the focal point (Supplementary Figs. 21 and 22). We then implemented a sophisticated graph structure to construct a 3D digital twin system with the following steps. Each detected 2D cross-sectional NCM contour and its adjacent pores were assigned distinct identifiers. These

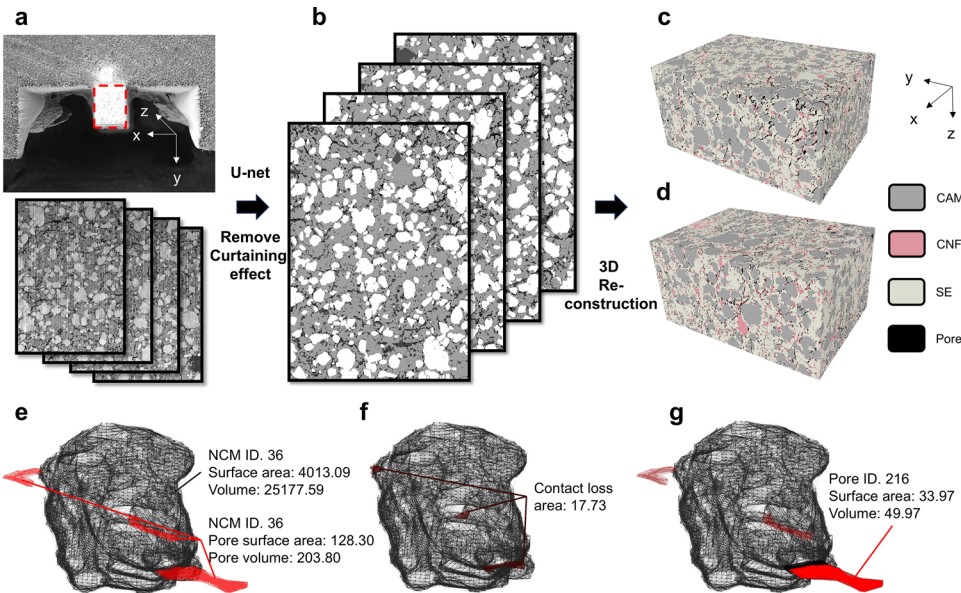

**Fig. 3 | Digital twin 3D reconstruction model from FIB−SEM images for quantitative microstructure analysis. a** Region of interest selected for sampling using FIB−SEM, followed by stacking of the obtained 2D images. Corresponding **b** four-phase segmentation results and reconstructed 3D images of the cathode composite having 28.28 × 39.08 × 55.82 µm³ volume for **c** bare NCM and d) LiDFP NCM.

Representative images of identification and quantitative information for **e** labeled cathode, **f** contact loss area, and **g** labeled pore. All the ASSB pellets were precycled in the range of 1.88–3.88 V (vs. In/InLi) under C/10 (1 C = 1.36 mA/cm²) at 25 °C with a stack pressure of 13 MPa.

identifiers, along with frame numbers and contour coordinates, were integrated into a graph structure (Supplementary Figs. 23 and 24). Using a template matching method, corresponding cathode particles were connected across consecutive frames, enabling 3D visualization of labeled particles (Supplementary Figs. 25 and 26). 3D object identifications were assigned, linking each 3D object to its adjacent pore IDs (Supplementary Fig. 27). The quantification process involved converting the pixel counts from 2D images within the 3D reconstruction to measure several key parameters: the surface area and volume of labeled NCM cathodes (Fig. 3e), the surface area and volume of pores surrounding specific cathode particles (Fig. 3e), the contact area between the cathode and adjacent pores (referred to as 'contact loss area'), as well as the surface area and volume of each adjacent pore (Fig. 3f, g). This comprehensive approach enables detailed analysis of the impact of contact loss and pore characteristics on the mechanical degradation and performance of the cathode composite, providing unprecedented insights into micro-scale behavior in ASSBs.

## Interplay of cathode reaction dynamics and microstructural evolution in ASSBs

To enhance our understanding of mechanical degradation in ASSB systems, we conducted a comprehensive analysis combining macro-scale porosity and tortuosity measurements with micro-scale statistical analysis of individual particles. This approach aimed to reveal detailed microstructural changes. We first validated our quantification method by comparing surface area, volume, and surface-area-to-volume ratio (uniformity) for 500 labeled cathode particles in both bare and LiDFP NCM samples (Fig. 4a–c). The identical distribution profiles observed for these parameters confirmed the reliability of our component identification and subsequent statistical comparisons. Initially, the porosity and ion tortuosity, which are considered conventional metrics for describing mechanical degradation, were elucidated. Contrary to previously noted phenomenological trends, where severe mechanical degradation results in detrimental effects on electrochemical performance such as cycle stability, it was observed that LiDFP-coated NCM exhibits 1.29 times higher porosity (7.16%) compared to bare NCM (5.54%). Ion tortuosity was 1.11 times longer in

LiDFP NCM (1.48) than in bare NCM (1.33) (Fig. 4d). These findings contrast with the results from 2D cross-sectioned SEM image (Supplementary Fig. 28), highlighting the limitations of 2D imaging in representing complex 3D microstructural evolution.

After confirming increased porosity in LiDFP NCM, we meticulously examined the contact loss area relative to the surface area of labeled cathode particles (Fig. 4e). Contact loss area typically increases with particle surface area, as larger particles undergo greater absolute volume change as the same SOC change[54]. Therefore, the contact loss ratio to the cathode surface area was derived from the slope value. This analysis revealed that LiDFP NCM exhibits nearly double the contact loss area compared to bare NCM (25.5% vs. 13.2%), based on a linear relationship considering equivalent total surface areas for each particle. The higher porosity (7.16%) in the LiDFP-coated cathode composite aligns with trends observed in contacted pore volume relative to cathode particle volume (Fig. 4d and Supplementary Fig. 29). Moreover, according to Eq. (8) (denoted in Supplementary Table 4)[55,56], current density is inversely proportional to tortuosity squared. Consequently, 3D spatial comparison consistently shows that the LiDFP NCM composite exhibits slightly lower ionic current density in localized areas (Supplementary Fig. 30).

$$J = \frac{\varepsilon}{\tau^2}\sigma\nabla E \tag{8}$$

Conventional wisdom regarding mechanical degradation would suggest that LiDFP NCM should exhibit more severe electrochemical deterioration over cycles compared to bare NCM, given its higher porosity, longer ionic tortuosity, and reduced active surface area; however, our observations strikingly contradict this expectation. This paradoxical behavior challenges traditional interpretations of degradation mechanisms in ASSBs. The unexpected performance of LiDFP NCM underscores the critical need for a more nuanced and comprehensive understanding of the intricate relationship between mechanical degradation and electrochemical performance in ASSB systems.

To elucidate microstructural changes, we evaluate the correlation between pore volume adjacent to each cathode particle and its contact

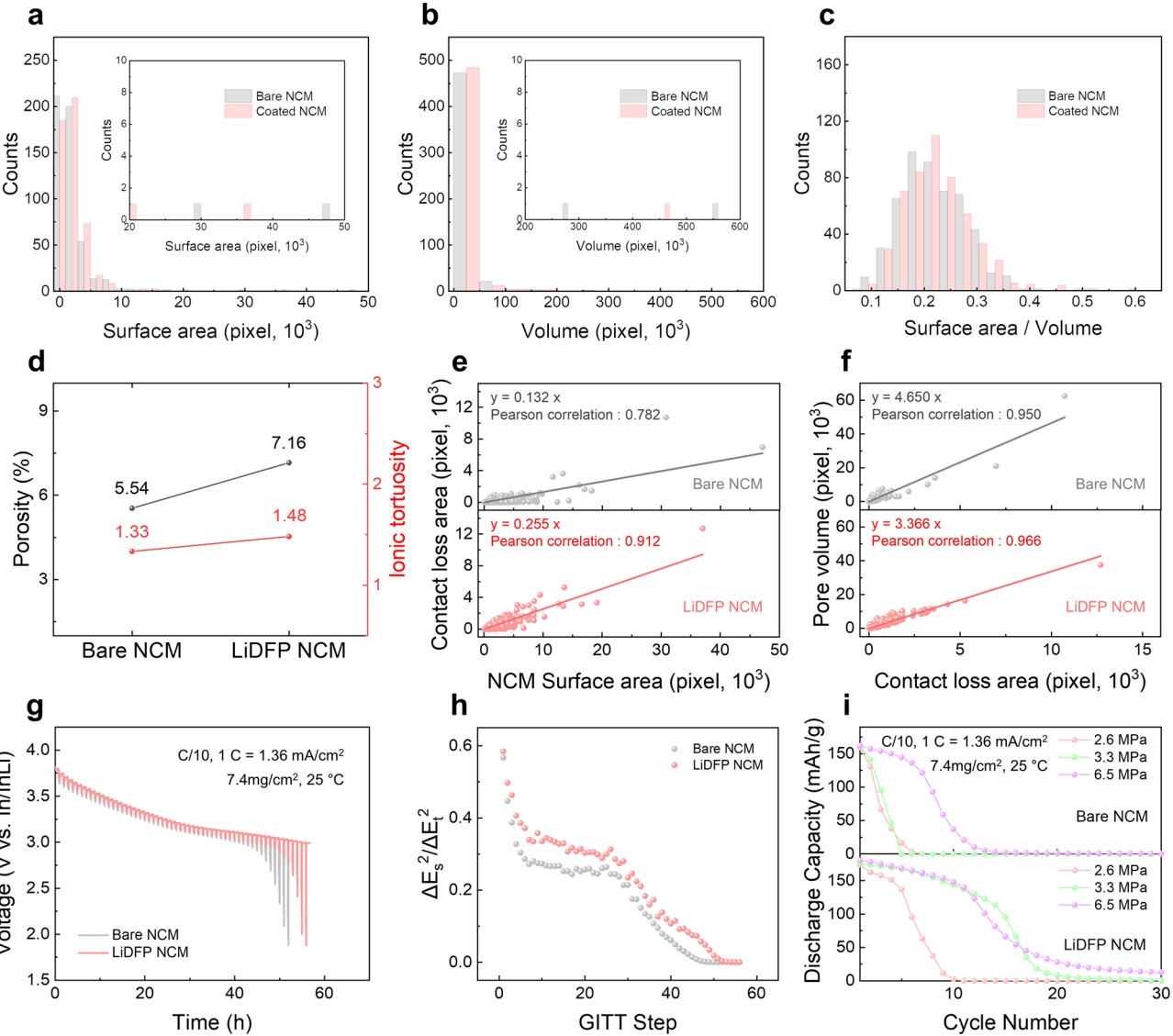

**Fig. 4 | Quantification of microstructural changes due to chemical degradation.** Histogram of labeled cathode particles for **a** surface area, **b** volume, and **c** surface area to volume ratio. **d** Comparative analysis of porosity and ionic tortuosity between bare NCM and LiDFP NCM within the reconstructed 3D composite electrode. **e** Comparison of cathode surface area versus contact loss area and **f** comparison of contact loss area versus adjacent pore volume for each labeled cathode particle in both bare NCM and LiDFP NCM. **g** GITT and OCV curves of In/InLi|LPSCl|NCM composite for bare NCM and LiDFP NCM during the first discharge cycle and corresponding **h** $\left(\frac{\Delta E_s}{\Delta E_t}\right)^2$ profiles for each GITT step of bare NCM and LiDFP NCM. **i** Electrochemical cycle performance of bare NCM and LiDFP NCM at C/10 (1C = 1.36 mA/cm²) rate and 25 °C under varying stack pressure of 2.6, 3.3, and 6.5 MPa with 7.4 mg/cm² active material mass loading.

loss area. Bare NCM exhibits significantly larger pore volumes and surface areas for equivalent contact loss areas, with 1.39 times greater volume and 1.29 times greater surface area compared to LiDFP NCM (refer to the slope value in Fig. 4f and Supplementary Fig. 31a). Moreover, LiDFP NCM necessitates the inclusion of a significantly greater number of pores compared to bare NCM at the equivalent pore volume on the cathode surface (Supplementary Fig. 31b). As demonstrated in Supplementary Videos 1 and 2, bare NCM, despite lower overall contact loss area, forms fewer but disproportionately large pores in concentrated regions. In contrast, LiDFP NCM generates a denser array of smaller, more uniformly distributed pores across the surface. These distinct microstructural behaviors reflect the influence of differing interfacial chemistries. In addition, Pearson's correlation coefficient ('r') analysis further supports these observations: LiDFP NCM maintains a high correlation coefficient (r > 0.9) between pore volume and contact loss area, whereas bare NCM shows a lower coefficient (r < 0.8) (Fig. 4e). This indicates superior homogeneity in

pore formation for LiDFP NCM, attributed to more uniform particle reactions, aligning with trends observed in in-situ XRD and TXM analysis.

The paradoxical behavior of LiDFP NCM – exhibiting a large contact loss area to the solid electrolyte yet demonstrating relatively homogeneous particle reaction compared to bare NCM—suggests a potential mechanism for maintaining active surface area through the surface coating layer, contrast to SEM observations. To investigate this phenomenon, we employed the galvanostatic intermittent titration technique (GITT) to evaluate the active surface area of both NCM and LiDFP NCM. It is important to note that in ASSB systems, the inherent nature of point contacts results in incomplete assurance of active surface area, presenting significant challenges in accurately determining lithium diffusivity via GITT[57]. Despite these limitations, we posit that a relative comparison of active surface area between bare and LiDFP NCM, as an assessment of mechanical degradation, is feasible using the following equations (detailed variable explanations are

provided in Supplementary Table 5). Note that the diffusion coefficient (D) obtained from GITT is an intrinsic bulk property. Because the coating is only a few nanometers thick and leaves the lithium-ion pathways in the bulk unchanged, the diffusivity should be identical for samples compared at the same state of charge (SOC). Consequently, variations in the voltage-relaxation profiles mainly indicate differences in the electrochemical contact area (S), rather than the intrinsic lithium diffusivity[58–60].

$$\left(\frac{\Delta E_s}{\Delta E_t}\right)^2 = k \times D^{GITT} \times S^2 \qquad (9)$$

$$k = \frac{\pi\tau}{4n_m^2 V_M^2} \qquad (10)$$

Given that $D^{GITT}$ is an intrinsic parameter dependent on the SOC, the value of $\left(\frac{\Delta E_s}{\Delta E_t}\right)^2$ at equilibrium state for the same SOC can serve as a relative indicator of active surface area. Hence, we employed GITT to comparatively analyze lithium-ion transport kinetics within the composite during discharge, evaluating differences between samples with and without the LiDFP coating (Fig. 4g)[61]. As shown in Fig. 1a, cathode particle charge capacity remains constant regardless of coating presence, ensuring GITT maintains identical SOC at each step during discharge (10-min discharge, 50-min relaxation; Supplementary Fig. 32). The 50-min relaxation period allows the system to reach thermodynamical equilibrium, mitigating lithium heterogeneity even at high DOH observed at the end of charging (Fig. 2b and Supplementary Fig. 33). According to Supplementary Fig. 34, LiDFP NCM typically exhibits a lower active surface area (~74.5% of total surface area) compared to bare NCM (~86.8%) for equivalent cathode surface area. Conventionally, this would result in lower $\left(\frac{\Delta E_s}{\Delta E_t}\right)^2$ values, as per Eq. (9). However, as illustrated in Fig. 4h, LiDFP NCM consistently demonstrates higher $\left(\frac{\Delta E_s}{\Delta E_t}\right)^2$ values across all SOC, suggesting a superior active surface area with the solid electrolyte for the LiDFP NCM composite. This unexpected finding implies that the active surface area can be maintained even with point contact between the coating layer and electrolyte, provided the coating layer maintains good contact with the cathode particles.

To further validate our findings, we evaluated the feasibility of ASSB operation under low pressure conditions. We conducted a comparative analysis of both LiDFP-coated and bare NCM cells under varying pressures (2.6, 3.3, and 6.5 MPa). Remarkably, despite exhibiting characteristics typically associated with severe mechanical degradation, such as higher porosity, increased ionic tortuosity, and a greater contact loss area, LiDFP NCM demonstrated stable operation at a low pressure of 3.3 MPa for over 10 cycles (Fig. 4i and Supplementary Fig. 35). Conversely, bare NCM required significantly higher pressure conditions, exceeding 6.5 MPa, to achieve comparable operational stability. This striking difference reveals an additional, crucial function of the surface coating layer beyond chemical degradation: it effectively maintains an active surface with the solid electrolyte even under low-pressure conditions. These results underscore the complex interplay between chemical and mechanical factors in ASSBs, highlighting the need for comprehensive understanding of chemo-mechanical behavior in these systems.

In summary, this study underscores a multi-length scale understanding of the chemical degradation at the solid electrolyte-cathode interface, its impact on particle ensemble reactions, and resulting microstructural changes. As illustrated in Fig. 5, we unveil the multifaceted roles of coating materials in the intertwined chemo-mechanical degradation mechanisms in ASSBs. In bare NCM, byproduct layer formation due to chemical degradation, coupled with randomized point contacts, significantly amplifies charge-transfer kinetics variability among cathode particles, exacerbating charge heterogeneity in bare NCM. Conversely, the LiDFP coating

suppresses interfacial decomposition, markedly improving charge uniformity and enhancing uniform reaction participation across cathode particles. Our statistical analysis of mechanical degradation further reveals that LiDFP NCM exhibits numerous small, uniformly distributed pores around the cathode surface, while bare NCM shows higher variability in pore distribution, accompanied by reaction heterogeneity among particles. This comprehensive micro-scale analysis highlights the need for a refined understanding of the interplay between the chemical and mechanical degradation in ASSBs to effectively comprehend the degradation processes. Both chemical and mechanical degradation at the cathode–electrolyte interface are driven by interrelated processes involving the cathode and the solid electrolyte, necessitating a coupled chemo-mechanical interpretation to fully understand interface degradation mechanism in ASSBs. Additionally, these findings also suggest a crucial role of the coating layer in maintaining lithium-ion diffusion to the cathode while preserving the interface contact, despite particle volume changes[62]. By elucidating these multi-scale interactions, our research establishes a vital foundation for developing strategies to mitigate degradation and enhance solid-state battery performance. This work paves the way for innovative approaches in the design and optimization of next-generation energy storage systems.

## Methods
### Material preparation
A sulfide-based argyrodite solid electrolyte, Li$_6$PS$_5$Cl (LPSCl) (Jeong Kwan Co., Ltd., Korea), and single-crystalline LiNi$_{0.6}$Mn$_{0.2}$Co$_{0.2}$O$_2$ (NCM cathode, BASF) were used in this study. A mechano-fusion-method-based mechanical powder precursor (Nob-Mini, Hosokawa Micron Co., Ltd., Japan) served as the dry-milling apparatus. This machine features a cylindrical vessel and a rotator. The vessel and rotator are positioned horizontally to ensure that the sample powder remains at the bottom of the inner circular wall, subjecting the powder to a strong shear force as it passes through the 1-mm gap between the vessel wall and rotor blade. The Li-salt additive, LiPO$_2$F$_2$ (LiDFP, 99.85% Chunbo Co., Ltd., Korea), was coated onto the NCM using the mechano-fusion coating method. The coating process involved dry milling at 2000 rpm for 10 min and at 4000 rpm for 30 min. Additionally, lithium foil (>99.8%, Wellcos Co., Korea) and indium foil (>99.8%, Wellcos Co., Korea) with Cu substrate (>99.8%, Wellcos Co., Korea) were placed in a 13-mm-diameter Teflon body and pressed under 295 MPa for 2 min to facilitate alloying.

### Cell assembly and electrochemical analysis
Full cells were assembled in an Ar-filled glove box with oxygen and moisture levels below 1 ppm. These cells were fabricated using the argyrodite electrolyte and a cathode composite, with a Li−In metal alloy serving as the counter electrode. The cathode composite was prepared by mixing the active material (65 mg), LPSCl (30 mg), and carbon nanofiber (5 mg) using a mortar and pestle. The cell pellet, comprising the cathode composite (15 mg), an electrolyte layer (120 mg), and a Li−In alloy with Cu substrate, was placed in a 13-mm-diameter Teflon body and pressed under 295 MPa for 2 min. A constant pressure of 4-Nm torque was applied uniaxially (stack pressure, ~13 MPa). Before galvanostatic cycling, the cells were cycled at C/10 for 4 cycles. Long-term cycling was conducted at C/4 (1 C = 1.36 mA/cm$^2$) at 25 °C within the voltage range of 1.88−3.88 V (vs. In/InLi) using a battery measurement system (WBCS 3000, WonATech). To evaluate the effect of the LiDFP coating on the rate performance, the full cells underwent cycling at various charge and discharge rates (C/10, C/4, C/2, 1.0C, 2.0C, 3.0C, and 0.1C). To ensure reproducibility of the observed trends, a set of at least two cells were examined for each electrochemical cell system.

The resistances of the full cells were assessed using electrochemical impedance spectroscopy (EIS) (Biologic, SP-300). For the

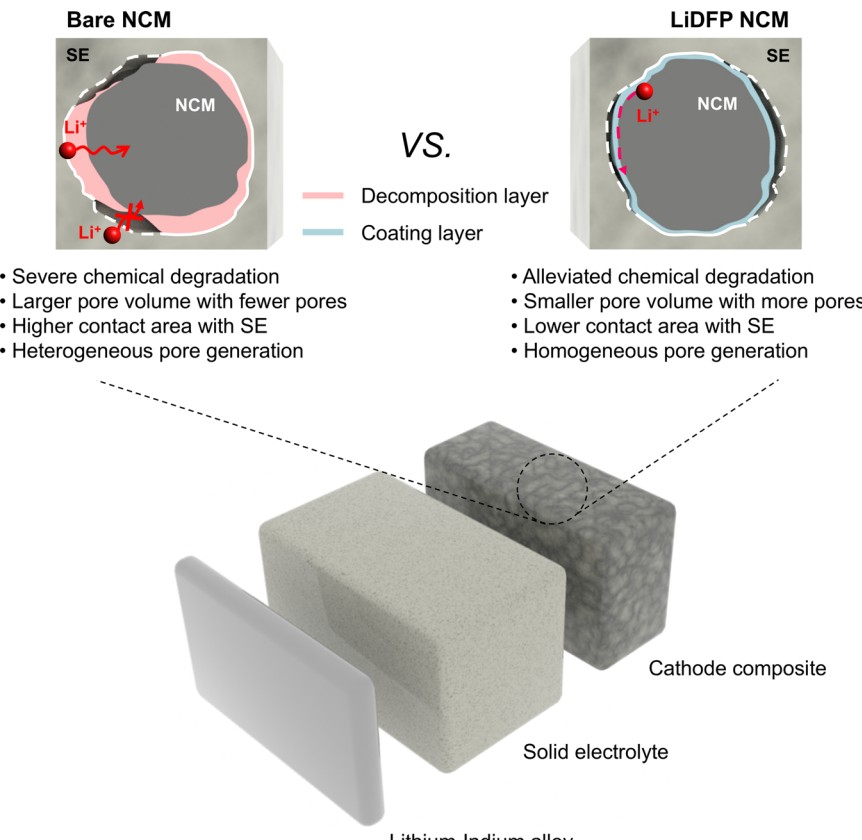

**Fig. 5 | Schematic illustration of microstructural changes in ASSBs induced by chemical degradation.** The LiDFP coating layer effectively mitigates chemical degradation at the cathode–SE interfaces, thereby encouraging more homogeneous cathode behavior. This results in smaller and more homogeneous pore formation while maintaining contact integrity between the cathode and coating layer, suggesting a potential pathway for lithium diffusion from the solid electrolyte to the cathode through the coating layer.

EIS measurements, potentiostatic EIS was applied at open-circuit voltage. The frequency range was set between 1 MHz and 100 mHz, with 10 points per decade in logarithmic spacing and a sinusoidal amplitude of 14.2 mV. Each EIS dataset was divided into high-frequency (HF), mid-frequency (MF), and low-frequency (LF) ranges using the MATLAB code DRTtools[37]. These frequencies correspond to the ionic resistance of the cathode composite, the CA|SE interface resistance, and the anode|SE interface resistance, respectively. The experimentally measured EIS data were fitted to $Z_{DRT}$ using the following Eq. (11):

$$Z_{DRT}(f) = R_{ohm} + \int_0^\infty \frac{\gamma(\tau)}{1 + i2\pi f\tau} d\tau = R_{ohm} + \int_{-\infty}^\infty \frac{\gamma(\ln\tau)}{1 + i2\pi f\tau} d\tau. \quad (11)$$

The Gaussian radial basis function and discrete parameters were obtained through regularized regression using the $2^{nd}$ order regularization derivative and regularization parameters of $10^{-3}$.

To elucidate the chemical stability in the presence of the Li-salt coating layer, In/InLi|SE|NCM full cells were charged to different cut-off voltages of 3.88 V (vs. In/InLi) and allowed to rest for 50 h. During this resting period, EIS analysis was performed every 27 min. The impedance spectra were measured by applying a 10-mV amplitude to the open-circuit voltage in the frequency range of 1 MHz to 0.1 Hz.

Cells for in-situ XRD (Smart Lab, RIGAKU) were constructed using an argyrodite electrolyte and a cathode composite, with a Li−In metal alloy as the counter electrode. The cell assembly included the cathode composite (20 mg), an electrolyte layer (200 mg), and the Li−In alloy, arranged in a 16-mm-diameter Teflon body and pressed under 297 MPa for 2 min. The pellet was then mounted on the in-situ XRD cell and subjected to a constant uniaxial pressure of 13 MPa. The cells were cycled at 0.2C (1C = 1.36 mA/cm²) at 25 °C within the voltage range of 1.88–3.88 V (vs. In/InLi) using a battery measurement system (SP-300, Biologic).

## Characterization of ASSBs

To minimize exposure to ambient air and ensure the accuracy of analytical results, all sample preparation steps were conducted in an argon-filled glove box with oxygen and moisture levels maintained below 1 ppm. Sample transfer to the analysis instruments was carried out either under inert argon atmosphere or via vacuum-sealed containers to prevent ambient contamination. The microstructure of the NCM cathode with the LiDFP coating was examined using HR-TEM (JEM-2100F, JEOL). Cross-sectional TEM analysis samples were prepared using FIB (NX2000, Hitachi). The presence of the LiDFP coating layer was confirmed using TOF-SIMS (TOF-SIMS 5, ION TOF GmbH) with a pulsed $Bi^{3+}$ ion beam (25 keV) in high current mode. After fabricating the In/InLi|LPSCl|NCM composite full cell using a 16-mm-diameter mold, the pellet was retrieved and placed onto an in-situ XRD holder. Utilizing Cu Kα radiation, the X-ray beam was directed at the cathode composite in reflection mode, focusing on the 18°–19° range to analyze the (003) peak, with measurements taken every 6 min during galvanostatic cycling[39]. XANES imaging experiments were performed using full-field transmission X-ray nano-imaging at beamline 7C at PLS-II in Pohang Accelerator Laboratory on the retrieved cathode composite after electrochemical testing. The typical FOV is approximately 273 μm, with an effective pixel size of approximately 135 nm at 8.33 keV (Ni K-edge)[40,63]. Energy-dependent images were acquired across the energy range at the Ni K-edge energy range at 1 eV 0.5–1 eV

intervals with an exposure time of 0.3 s for each image, and aligned using the in-house beamline software.

For the FIB 3D images, ASSB pellets retrieved from a 13-mm-diameter Teflon body and pressed under a uniaxial stack pressure of approximately 13 MPa were subjected to FIB-SEM imaging (PP3010, Quorum). These pellets were positioned at a 36° tilt to ensure that the cathode composite faced the ion beam perpendicularly. Initially, the ion beam created islands around the target area, and an SEM image was taken to capture the 2D morphology of the cathode composite. Subsequently, thin slices of 97.52 nm, with a pixel size of 46.52 nm, were sequentially removed from the cross-section using the ion beam. Each removal was followed by SEM image capture, repeated multiple times. To ensure the fidelity of 3D digital twin reconstruction, we evaluated potential artifacts arising from differential ion etching of component phases—particularly the mechanically soft LiDFP coating layer. Comparative FIB–SEM imaging of fresh-state cathode composites (as shown in Supplementary Fig. 3) revealed no significant differences in pore contrast or morphology. Additionally, the voxel size of the reconstruction (46.5 nm) substantially exceeds the LiDFP coating thickness (5–10 nm), minimizing the likelihood of artificial porosity resulting from coating-specific etching. These validations confirm that the presence of LiDFP does not compromise the accuracy of porosity or connectivity analysis in the reconstructed model.

### Image processing and reconstruction

The difference between the pixel size and slice thickness was adjusted using the nearest function, aligning both to 46.52 nm. Images were cropped to $840 \times 1200 \times 608$ pixels to fit the region of interest, with a domain size of $39.08 \times 55.82 \times 28.28$ μm. The wavelet-FFT function was employed to mitigate the curtaining effect, and flickering correction was applied to address differences in gray values along the electrode-thickness direction (y-axis). To further eliminate residual noise, the non-local-means filter was used for image filtering. These processes of image alignment, cropping, and filtering were conducted using various plugins in ImageJ, Avizo, and the ImportGeo-Vol module of Geo-Dict2024 (Math2Market GmbH). For image segmentation, we adapted the convolutional neural network-based U-net algorithm, commonly utilized in biomedical image segmentation, for the composite cathodes of ASSBs. Through U-net training, we segmented NCM, LPSCl, CNF, and Pore, constructing a digital twin structure from the results. This digital twin structure was then used for simulation and calculation. To evaluate the segmentation accuracy of the U-net-based 3D reconstruction, we adopted the commonly used validation method—comparing the segmented volume fractions with experimentally input values based on material densities. These values show excellent agreement, with discrepancies less than 6% between the segmentation-derived and actual experimental volume ratios, confirming the reliability of the digital twin model. The electronic and ionic current density maps were calculated using Ohm's law, with $\Delta V = 1$ V under Dirichlet boundary conditions, and this calculation was implemented using the explicit jump solver of the ConductoDict module in GeoDict2024.

### Data availability

All data that support the findings of this study are presented in the Manuscript, Supplementary Information and Supplementary Videos, or are available from the corresponding author upon request. Source data are provided with this paper.

### Code availability

The basic code structure for the digital twin system used in this study is available at Zenodo https://doi.org/10.5281/zenodo.16924323. For inquiries regarding additional data and code, please contact Prof. Donghyuk Kim and Prof. Sung-Kyun Jung (dkim@unist.ac.kr and naecard@snu.ac.kr).

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

## Acknowledgements

This work was supported by the Technology Innovation Program (20017836, Inorganic material-based new electrolyte additive for secondary battery) funded by the Ministry of Trade, Industry & Energy (MOTIE, Korea), a National Research Foundation of Korea (NRF) grant funded by the Korea government (MSIT) (No. 2022R1C1C1006575), the POSCO Science Fellowship of POSCO TJ Park Foundation, the Nano & Material Technology Development Program through the National Research Foundation of Korea(NRF) funded by Ministry of Science and ICT(RS-2024-00406724), the National Research Foundation (NRF) funded by the Korean government (MSIT) (No. RS-2024-00440681), Korea Institute for Advancement of Technology (KIAT) grant funded by the Korea Government(MOTIE) (RS-2024-00419413, HRD Program for Industrial Innovation), the National Research Foundation of Korea (NRF) grant funded by the Korea government (MSIT) (No. RS-2023-00261543) and the Chunbo Company.

## Author contributions

C.P. and S.-K.J. designed the project and wrote the manuscript. C.P., Juho L., and Jinsoo K. conceived the coating synthesis. Surface analysis was performed by C.P. and Jiwon K. Electrochemical tests and characterizations, including in-situ XRD and ex-situ TXM, were conducted by C.P., J.C., Juho L. and E.L. The concept of the degree of heterogeneity (DOH) was suggested by C.P., J.C., and S.-K.J. Program coding and comments for the ex-situ TXM analysis were provided by G.L., S.J., Jun L., and T.K. Constructive advice regarding the in-situ XRD and ex-situ TXM analyses was provided by J.H. The digital twin 3D reconstruction of the FIB-SEM images was conducted by H.-J.K., with quantitation analysis on labeled cathode particles retrieved from 3D reconstructed images provided by S.P., Y.K., and D.K. J.H. and D.K. also provided critical feedback on the project. S.-K.J. supervised all aspects of the research. All the authors discussed the results and contributed to the manuscript.

## Competing interests

The authors declare no competing interests.

## Additional information

**Chanhyun Park** [1,2,15], **Jingyu Choi**[1,3,15], **Seojoung Park** [1,4,15], **Hyeong-Jong Kim** [1,3], **Yunseo Kim** [1], **Gukhyun Lim**[5], **Juho Lee** [1,6], **Eunryeol Lee** [7,8,9], **Sugeun Jo** [10], **Jiwon Kim**[3], **Jinsoo Kim** [6], **Jun Lim** [10], **Taeseok Kim**[11], **Jihyun Hong** [5] ✉, **Donghyuk Kim** [1] ✉ & **Sung-Kyun Jung** [3,12,13,14] ✉

[1]School of Energy and Chemical Engineering, Ulsan National Institute of Science and Technology (UNIST), Ulsan, Republic of Korea. [2]Institute of Physical Chemistry and Center for Materials Research (ZfM), Justus-Liebig-University Giessen, Giessen, Germany. [3]Department of Materials Science and Engineering, College of Engineering, Seoul National University, Seoul, Republic of Korea. [4]Samsung SDI R&D center, Suwon-si, Gyeonggi-do, Republic of Korea. [5]Department of Battery Engineering, Graduate Institute of Ferrous & Eco Materials Technology, Pohang University of Science and Technology (POSTECH), Pohang-si, Gyeongsangbuk-do, Republic of Korea. [6]Ulsan Advanced Energy Technology R&D Center, Korea Institute of Energy Research, Ulsan, Republic of Korea. [7]Department of Materials Science and Engineering, University of California Berkeley, Berkeley, CA, USA. [8]Materials Sciences Division, Lawrence Berkeley National Laboratory, Berkeley, CA, USA. [9]Department of Chemical Engineering, Chungbuk National University, Cheongju, Chungbuk, Republic of Korea. [10]Pohang Accelerator Laboratory, Pohang University of Science and Technology (POSTECH) 80, Pohang-si, Gyeongsangbuk-do, Republic of Korea. [11]Company Lomin, JBI Building, Seoul, Republic of Korea. [12]School of Transdisciplinary Innovations, Seoul National University, Seoul, Republic of Korea. [13]Research Institute of Advanced Materials (RIAM), Seoul National University, Seoul, Republic of Korea. [14]Institute for Battery Research Innovation, Seoul National University, Seoul, Republic of Korea. [15]These authors contributed equally: Chanhyun Park, Jingyu Choi, Seojoung Park. ✉e-mail: jhong@postech.ac.kr; dkim@unist.ac.kr; naecard@snu.ac.kr

