## [Transparent Peer Review file · Nature Communications]

Interfacial Chemistry-Driven Reaction Dynamics and Resultant Microstructural Evolution in Lithium-based All-Solid-State Batteries

Corresponding Author: Professor Sung-Kyun Jung

Version 0:

Reviewer comments:

Reviewer #1

(Remarks to the Author)

In this study, the authors investigated the effect of chemical degradation at the interface of single crystal NCM cathode/solid electrolyte on the reaction behavior and microstructural evolution of cathode particles in composite cathodes of sulfide-based ASSBs. Through a series of advanced characterization techniques, the authors confirmed that the use of lithium difluorophosphate (LiDFP) to inhibit chemical degradation can enhance the reaction uniformity between cathode particles and homogenize mechanical degradation. Overall, this study presents some interesting phenomena. However, there are still several important issues that require further investigation.

1. The focus of this study is on sulfide-based solid electrolytes. The introduction should clearly state the rationale and innovation behind the selection of sulfide electrolytes, especially considering that oxide solid electrolytes generally exhibit more stable mechanical and chemical properties.
2. While this study concentrates on single-crystal NCM cathode materials, conventional battery materials are typically polycrystalline. The authors should address the practical significance of their research in this context.
3. The concept of studying interface behavior by coating ion conductors on the cathode surface is not novel. The authors need to articulate the core innovation of their work.
4. For interface degradation, could more intuitive characterization methods, such as transmission electron microscopy (TEM), be employed?
5. Is the interface change during the reaction process a result of the cathode, or is it influenced by factors such as electrolyte decomposition?

(Remarks on code availability)

Reviewer #2

(Remarks to the Author)

In this paper titled "Interfacial Chemistry-Driven Reaction Dynamics and Resultant Microstructural Evolution in All-Solid-State Batteries," the authors examine the complex chemo-mechanical degradation occurring in composite cathodes of sulfide-based ASSBs. To visualize the evolution of the mechanical properties of composite cathodes, they propose a novel and intelligent digital twin 3D reconstruction model based on FIB-SEM images for quantitative microstructure analysis. This approach paves the way for revealing the mechanical properties of the solid-solid contact in ASSBs. However, the analysis of the complex interplay of chemical and mechanical behaviors at the cathode/solid-electrolyte interface is not rigorous. Adjustments or supplements are required to support the hypothesis of this manuscript. This article can be considered for publication in Nature Communications after major revisions. Here are the issues you may refer to:

1. The authors employed the mechano-fusion method to synthesize the LiDFP coating NCM cathode. Supplementary Fig. 1a confirmed a homogeneous and thin LiDFP-based film (approximately 10 nm thick) on the cathode surface. However, the morphological properties of the LiDFP and LiDFP NCM on the micrometer scale should also be provided so that the changes can be simultaneously comprehended after mechanical mixing.
2. The electrochemical properties shown in Figures 1. e and 1. f are inconsistent. Reproducible raw data should be provided to clarify the properties' differences between the modified and pristine samples, ensuring self-consistency within this

manuscript. The same concern is also present in Figure 4. i.

3. The basis for Equation 5 in the manuscript concerning the chemical reactions of the Li₆PS₅Cl electrolyte and cathode and the volume change ratio needs to be proven or cited. Additionally, there is a minor clerical error in the writing of Equation 5. Please review the entire manuscript for clarity and accuracy!
4. The reasons for the leaning toward a lower angle of Ni K-edge X-ray absorption near edge structure spectrum (XANES) at 0% SOC after LiDFP coated should be valued. This phenomenon may be caused by the valence distribution of nickel ions, the evolution of the surface structure of the NCM cathode materials, and other contributing factors. Additionally, it is essential to consider the effect of the electronegativity of the ligand surrounding Ni and the coordination structure. This aspect should be included in the elaboration. If not addressed, this could result in a logical disconnect.
5. The assumption of "Despite these limitations, we posit that a relative comparison of active surface area between bare and LiDFP NCM, as an assessment of mechanical degradation, is feasible using the following equations" lacks a degree of rigor. The active contact area and the ion diffusion coefficient have previously been interlinked. Therefore, a more thorough explanation may be warranted.
6. The enhanced performance of LiDFP NCMs primarily stems from their capacity to maintain active contact surfaces and facilitate lithium-ion diffusion. Is it possible to effectively quantify how the active surface contributes to the performance of anode materials?
7. 3D reconstruction using FIB-SEM effectively facilitates quantitative microstructural analysis. However, the use of e.g. U-net diffusion models for image optimization in this technique may result in variations in microregion pixel points. Do the authors have any comments on the error evaluation of the model?

(Remarks on code availability)

The code provided in the manuscript does not facilitate repeated runs, as it is not accompanied by corresponding example data. From the code and the results displayed by Jupyter notebook, the program seems to be no problem.

Reviewer #3

(Remarks to the Author)

Major revision

All-solid-state batteries based on sulfide electrolytes attained enormous attentions owing to their high energy density and high intrinsic safety. Nevertheless, the complex microstructural evolution including pore formation and contact failure originated from chemical/electrochemical instabilities of cathodes/solid electrolytes interfaces and breath effect of cathodes, hinder the enhancement for battery performance. Herein, the authors investigate the effect of interfacial attenuation on electrochemical properties of cathodes and the mechanism of microstructure evolution, by constructing a well-defined model system (LiDFP@NCM622). The whole work is impressive and valuable. But there are still some problems to be addressed before the consideration for publication.

1. The well-defined model has been constructed with uniform LiDFP coating layer onto NCM. There are clear observations in Fig. S1a, but the elemental distributions of cross-sectioned STEM image are still in need for the precise distinction of heterojunction structure.

2. The detailed assembly processes of all-solid-state batteries for in-situ XRD testing of should be stated. At the same time, in the XRD test, considering that the battery contains a variety of components, the signals representing other components besides cathode actually exist, and how the authors distinguish and exclude them should be indicated. Relevant theoretical analysis and systematic experimental demonstration should be given.

3. Information such as mass loading of active materials and test temperature of the all-solid-state batteries mentioned in this manuscript should be given and labeled in figures for readers' better understanding.

4. The in/ex-situ characterizations of the composite cathodes of all-solid-state batteries described in this manuscript are impressive. However, considering the air sensitivity of sulfide electrolytes, how does the authors eliminate the above interference during characterizations to ensure the validity of data? Do the current test methods meet the requirements of air tightness? Relevant instructions should be given.

5. During FIB-SEM analysis, the thin slices are removed by ionic etching, considering the different properties of components in composite cathodes, especially the LiDFP coating layer with low Yong's modulus as authors mentioned, does etching rates of them will be different? And does this eventually create excess porosity during the reconstruction of digital twinned 3D model? The detailed verifications and descriptions need to be added.

(Remarks on code availability)

The code has been viewed, which guarantees the reproducibility of cathodic synthesis, batteries assembly, advanced characterizations and performance testing.

Version 1:

Reviewer comments:

Reviewer #1

(Remarks to the Author)

The authors have perfectly addressed my issues.

(Remarks on code availability)

Reviewer #2

(Remarks to the Author)

The newly submitted manuscript shows that the authors have carefully made sensible revisions to the article in response to the comments. As for the error evaluation of the model, the accuracy evaluation used for image segmentation is effective, enabling subsequent microstructural quantification. The revised manuscript is acceptable for Nature Communications. However, one mistake in Equation 7 should be corrected before next step.

The error : $7Li_6PS_6(6?)Cl + 24LiNiO_2 \rightarrow 7LiCl + 7S + 8Ni_3S_2 + 7Li_3PO_4 + 5Li_2SO_4 + 7Li_2S$.

(Remarks on code availability)

Reviewer #3

(Remarks to the Author)

Based on a thorough review and evaluation of the author's responses to the reviewers, I consider this work to possess significant reference value in the field of all-solid-state batteries and to demonstrate notable novelty compared to published literature.

Furthermore, the robust experimental data and theoretical analysis presented convincingly support the paper's logical framework and conclusions. The scientifically sound experimental methodologies employed align with prevailing standards in the field. Sufficient experimental details are provided to ensure experimental replicability.

Therefore, I recommend its publication in Nature Communications.

(Remarks on code availability)

Response to Referees Letter

Title: Interfacial Chemistry-Driven Reaction Dynamics and Resultant Microstructural Evolution in All-Solid-State Batteries

Authors: Chanhyun Park[†], Jingyu Choi[†], Sejoung Park[†], Hyeong-Jong Kim, Yunseo Kim, Gukhyun Lim, Juho Lee, Eunryeol Lee, Sugeun Jo, Jiwon Kim, Jinsoo Kim, Jun Lim, Taeseok Kim, Jihyun Hong*, Donghyuk Kim*, and Sung-Kyun Jung*

Manuscript Number: NCOMMS-25-10677-T

Reviewer comments:

Reviewer #1 (Remarks to the Author):

Comments:

In this study, the authors investigated the effect of chemical degradation at the interface of single crystal NCM cathode/solid electrolyte on the reaction behavior and microstructural evolution of cathode particles in composite cathodes of sulfide-based ASSBs. Through a series of advanced characterization techniques, the authors confirmed that the use of lithium difluorophosphate (LiDFP) to inhibit chemical degradation can enhance the reaction uniformity between cathode particles and homogenize mechanical degradation. Overall, this study presents some interesting phenomena. However, there are still several important issues that require further investigation.

Author reply:

We appreciate the reviewer for careful review on our manuscript and delivering helpful comments. In the following, we tried our best to address the reviewer's concern and clarify our argument with additional experiment. We hope that the manuscript is now ready to be published in *Nature Communications*. We reproduced the comments of the reviewer, responded to the comments and suggestions.

Comments:

1. The focus of this study is on sulfide-based solid electrolytes. The introduction should clearly state the rationale and innovation behind the selection of sulfide electrolytes, especially considering that oxide solid electrolytes generally exhibit more stable mechanical and chemical properties.

Author reply:

We sincerely thank the reviewer for this insightful and important comment. We fully agree that oxide-based solid electrolytes—such as garnet-type $\text{Li}_7\text{La}_3\text{Zr}_2\text{O}_{12}$ (LLZO)—are well known for their outstanding chemical and mechanical stability across a wide range of conditions. These properties have made them promising candidates for solid-state battery systems. However, the core objective of our study is to analyze chemically induced behaviors in cathode composites, such as reaction dynamics and the associated microstructural evolution. From this perspective, we determined that sulfide-based solid electrolytes provide a more suitable experimental platform for our investigation, for several important reasons. Although oxide solid electrolytes offer superior chemical stability and are thermodynamically less prone to decomposition, they often require high-temperature annealing or sintering steps (typically $>700\text{ }^\circ\text{C}$) to form sufficient contact with active cathode particles [A. Banerjee, *et. al.*, *Chem. Rev.* 2020, 120 6878-6933]. This annealing step, while improving contact, frequently results in the formation of resistive interfacial layers, including lithium carbonate (Li_2CO_3), lithium zirconate (Li_2ZrO_7), and other mixed-phase byproducts such as LaCoO_3 , depending on the cathode composition and electrolyte interaction [S. Ohta, *et. al.*, *J. Power Sources* 2012, 202, 332-335]. Further interdiffusion between cathode and SE was also verified by Vardar *et al.* where they found interdiffusion of Co, La, and Zr across the interface after annealing the LLZO-LCO composite at $500\text{ }^\circ\text{C}$, though no interfacial elemental distribution was found

without annealing [G. Vardar, *et. al.*, *Chem. Mater.* 2018, 30, 6259-6276] as shown in **Fig. R1**. As a result, although some oxide electrolytes have a higher decomposition energy than sulfides, they paradoxically tend to exhibit higher interfacial resistance due to the formation of ionically blocking layers at the cathode interface with annealing step [A. Banerjee, *et. al.*, *Chem. Rev.* 2020, 120, 6878-6933]. These insulating products inhibit both charge transfer and ionic conductivity, ultimately elevating the cathodic overpotential and impeding reaction uniformity which hinders sophisticated understanding for chemically induced cathode behavior. In addition, the inherent mechanical brittleness of oxide electrolytes poses further challenges. Materials such as LLZO often experience cracking under modest pressure, complicating the mechanical interpretation of cathode–electrolyte interaction [K. J. Kim, *et. al.*, *Adv. Energy Mater.* 2021, 11, 2002689]. This structural rigidity limits our ability to isolate the effects of chemically driven pore evolution or stress accommodation within the cathode composite—a crucial aspect when studying long-term degradation in ASSBs.

In contrast, sulfide-based solid electrolytes, such as $\text{Li}_6\text{PS}_5\text{Cl}$, offer several key advantages that align well with the goals of our study. First, they can form intimate contact with cathode particles without high-temperature sintering, preserving interfacial chemistry and reducing the formation of interphases. Sulfides are also highly compatible with a range of nanometer-thin coatings, including LiNbO_3 , $\text{Li}_4\text{Ti}_5\text{O}_{12}$, and LiAlO_2 , which are frequently employed to stabilize interfacial reactivity under practical cycling conditions [Y. Seino, *et. al.*, *J. Power Sources* 2011, 196, 6488-6492; K. Okada, *et. al.*, *Solid State Ion.* 2014, 255, 120-127]. In our prior work, we also introduced a thermal-synthesis-free coating process that prevents phase evolution or undesired chemical mixing at the cathode interface [C. Park *et. al.*, *Adv. Energy Mater.* 2023, 13, 2203861]. Equally important, the relatively soft mechanical nature of sulfide electrolytes enables more direct observation of the coupling between interfacial reactivity and microstructural evolution. In particular, it allows us to track how chemically modulated reaction dynamics lead to morphological changes such as surface pore formation—a central focus of our investigation [K. J. Kim, *et. al.*, *Adv. Energy Mater.* 2021, 11, 2002689]. Moreover, sulfide electrolytes typically exhibit higher ionic conductivity and lower gravimetric density than many oxide counterparts, which are desirable characteristics for the practical implementation and energy density optimization of all-solid-state batteries (ASSBs) [Y. Wang, *et. al.*, *Nat. Mater.* 2015, 14, 1026; C. Park *et. al.*, *Adv. Energy Mater.* 2023, 13, 2203861]. In summary, sulfide-based solid electrolytes offer critical advantages in terms of low-temperature processability, mechanical compliance, interfacial tunability, and experimental transparency, making them an ideal platform for the mechanistic exploration of chemically driven degradation and interfacial evolution than oxide based solid electrolytes. These benefits directly support the objectives of our study and enable a clearer understanding of how interfacial chemistry governs electrochemical behavior and structural transformation in ASSBs.

Fig. R1. a) XRD patterns of an LLZO pellet, LCO deposited on LLZO, and LCO–LLZO heated at 500 °C with the corresponding phases. FIB-milled sample of the LCO and LLZO interface b) before and c) after annealing with the corresponding EDX line scans (of Co, La, and Zr). Co, La, and Zr were shown to diffuse after annealing.

Referred by [G. Vardar, et. al., Chem. Mater. 2018, 30, 6259-6276]

We have clarified this rationale in the revised manuscript to ensure that the innovation and appropriateness of our material selection are clearly articulated to the reader as below:

Original text (Page 5, line 10)

In this study, we aimed to investigate the impact of chemical degradation at the cathode/solid-electrolyte interface on cathode-particle reaction behavior and microstructural changes in sulfide-based ASSBs. We employed lithium difluorophosphate (LiDFP , LiPO_2F_2) as a model coating material to suppress chemical degradation²⁷, analyzing reaction uniformity and microstructural changes in composite electrodes with and without the coating.

Revised text (Page 5, line 10)

In this study, we aimed to investigate the impact of chemical degradation at the cathode/solid-electrolyte interface on cathode-particle reaction behavior and microstructural changes in sulfide-based ASSBs. To elucidate the intertwined chemo-mechanical behavior of ASSBs, we utilized a model system comprising single-crystalline cathodes and sulfide-based solid electrolytes ($\text{Li}_6\text{PS}_5\text{Cl}$), which offer significant advantages. While polycrystalline cathodes are widely used for their high tap density and cost advantages, they

suffer from intergranular cracking and internal pore formation during cycling and pressing, leading to mechanical degradation and electrochemical isolation of cathode particles—both of which adversely affect ASSB performance.^{21,22} In contrast, single-crystal cathodes, free from intergranular cracking, offer a more crack-resistant architecture that mitigates these issues and enables clearer observation of chemically driven interfacial effects.²³ Furthermore, oxide-based electrolytes must undergo high-temperature annealing, which often densifies the interface and produces undesirable secondary phases at the cathode.^{24,25} By contrast, sulfide solid electrolytes can be processed at low temperatures, enabling a conformal and intimate contact with the cathode particles. Their relatively compliant mechanical nature minimizes artificial fracture or densification effects, allowing more accurate observation of chemically induced interfacial and morphological changes.²⁶ We employed lithium difluorophosphate (LiDFP, LiPO₂F₂) as a model coating material to suppress chemical degradation²⁷, analyzing reaction uniformity and microstructural changes in composite electrodes with and without the coating.

Added references

24. Vardar, G. *et al.* Structure, Chemistry, and Charge Transfer Resistance of the Interface between Li₇La₃Zr₂O₁₂ Electrolyte and LiCoO₂ Cathode. *Chem. Mater.* **30**, 6259–6276 (2018).

25. Ohta, S. *et al.* Electrochemical performance of an all-solid-state lithium ion battery with garnet-type oxide electrolyte. *J. Power Sources* **202**, 332–335 (2012).

26. Kim, K. J. *et al.* Solid-State Li–Metal Batteries: Challenges and Horizons of Oxide and Sulfide Solid Electrolytes and Their Interfaces. *Adv. Energy Mater.* **11**, 2002689 (2021).

Comments:

2. While this study concentrates on single-crystal NCM cathode materials, conventional battery materials are typically polycrystalline. The authors should address the practical significance of their research in this context.

Author reply:

We sincerely thank the reviewer for this insightful question. While polycrystalline layered oxide cathodes—such as poly-NCM and poly-NCA—remain the dominant choice in commercial lithium-ion batteries due to their favorable processability and high tap density, we deliberately employed single-crystal NCM in this study to construct a platform capable of decoupling chemically induced interfacial behavior from complex structural artifacts.

Polycrystalline cathodes are known to suffer from intragranular and intergranular cracking during electrochemical cycling, particularly under deep cycling or elevated mechanical strain. These fractures create pathways for parasitic side reactions, disrupt particle connectivity, and promote the formation of electrochemically isolated active material (AM) domains, ultimately resulting in impedance buildup and capacity fade. As shown in Figure R2, such fracture phenomena compromise structural integrity and facilitate interfacial instability [S. Lee *et al.*, *J. Mater. Chem. A* 2024, 12, 26244–26252]. Moreover, the localized volumetric strain driven by side reactions with sulfide-based solid electrolytes (e.g., LPSCl, LPSX) has been shown to further accelerate the pulverization of polycrystalline secondary particles, particularly in poly-

NCA systems [Y. Han *et. al.*, *Adv. Energy Mater.* 2021, 11, 2100126]. Recent studies also demonstrate that improved crack resistance of cathode particles can directly impact oxygen release from the NCM lattice, which in turn alters the nature and extent of the SEI/CAM decomposition layer [S. Payandeh *et. al.*, *Adv. Mater. Interfaces* 2023, 10, 2201806]. These observations emphasize how structural degradation at the particle level interferes with the interpretation of interfacial chemistry, thereby complicating efforts to unravel the chemically driven mechanisms in all-solid-state batteries (ASSBs).

Fig. R2. Schematic illustration of particle fracture in polycrystalline NMC materials.

Referred by [G. Qian *et. al.*, *Energy Storage Mater.* 2020, 27, 140-149]

In our work, the use of single-crystal NCM allows us to eliminate these grain boundary–induced complications and instead focus on how interfacial reactivity—modulated by the presence or absence of coating layers—affects reaction uniformity and pore formation on the cathode surface. The simplified microstructure of single-crystal particles provides a clean model system for isolating the influence of chemical factors without the interference of cracking-related effects.

Importantly, we believe this material choice is not merely a model simplification, but also represents a practically relevant direction for next-generation ASSB design. In conventional liquid systems, electrolyte infiltration can compensate for fractured pathways, but in solid-state systems, intergranular cracks lead to irreversible ionic isolation. This is especially problematic for micron-sized polycrystalline particles, where networks of grain boundaries and inherent porosity can intensify stress accumulation, leading to charge inhomogeneity and performance loss. These effects are often compounded by anisotropic lithium transport and phase separation, resulting in persistent volume expansion, local contraction, and eventual microcrack formation throughout the active material as shown in **Fig. R3** [S. Lou, *et. al.*, *Nat. Commun.* 2020, 11, 1-10]. As a result, particle-level isolation from cracking presents a critical risk to electrochemical performance. Single-crystal cathodes, with their enhanced mechanical robustness, minimize the formation of isolated AM regions and preserve conductive networks throughout cycling [G. Qian *et. al.*, *Energy Storage Mater.* 2020, 27, 140-149]. This practical relevance is further supported by **Fig. R4**, which illustrates the differences in pulverization behavior, morphology after pressing, and schematic particle characteristics between polycrystalline and single-crystal NCM. These results confirm the superior mechanical tolerance and structural stability of the

single-crystal system [H. Cha *et. al.*, *Adv. Mater.* 2020, 32, 2003040]. In addition, recent literature increasingly supports the shift toward single-crystal-based cathode architectures for solid-state batteries due to their favorable interfacial conformity, high volumetric density, and superior long-term cycling behavior

Fig. R3. Structure and SOC distribution of the cycled cathode particles. a) Comparison of the cycling stability of NCM electrodes in the ASSLBs and LELBs at 0.5 C after activation. b, c) Typical SEM images of the cycled NCM electrodes in the ASSLBs, scale bar, 2 μm , 3 μm . d) 2D TXM-XANES mapping of cycled NCM particle in the ASSLBs, and e) diagram of the effects of cracks on lithium transportation. f) 2D TXM-XANES mapping of cycled NCM particle in the LELBs, and g) diagram of the slight effects of microcracks on lithium transportation. Scale bar, 10 μm . Referred by [S. Lou, *et. al.*, *Nat. Commun.* 2020, 11, 1-10].

Fig. R4. Characterization of the single crystal-NCM. a) Pellet density of polycrystalline-NCM and single crystal-NCM. b, c) SEM images of the polycrystalline -NCM b) and single crystal -NCM c) after pressing at 45 MPa. Schematics of the characteristics of d) polycrystalline -NCM e) and single crystal -NCM. Referred by [H. Cha *et. al.*, *Adv. Mater.* 2020, 32, 2003040]

We therefore believe our work provides not only a fundamental mechanistic insight into interfacial heterogeneity but also contributes to the emerging design paradigm for high-performance cathodes in ASSBs.

We have revised the manuscript to reflect this rationale and to emphasize the dual value of our material choice in both experimental design and practical relevance as below:

Original text (Page 5, line 10)

In this study, we aimed to investigate the impact of chemical degradation at the cathode/solid-electrolyte interface on cathode-particle reaction behavior and microstructural changes in sulfide-based ASSBs. We employed lithium difluorophosphate (LiDFP, LiPO₂F₂) as a model coating material to suppress chemical degradation²⁷, analyzing reaction uniformity and microstructural changes in composite electrodes with and without the coating.

Revised text (Page 5, line 10)

In this study, we aimed to investigate the impact of chemical degradation at the cathode/solid-electrolyte interface on cathode-particle reaction behavior and microstructural changes in sulfide-based ASSBs. **To elucidate the intertwined chemo-mechanical behavior**

of ASSBs, we utilized a model system comprising single-crystalline cathodes and sulfide-based solid electrolytes (Li₆PS₅Cl), which offer significant advantages. While polycrystalline cathodes are widely used for their high tap density and cost advantages, they suffer from intergranular cracking and internal pore formation during cycling and pressing, leading to mechanical degradation and electrochemical isolation of cathode particles—both of which adversely affect ASSB performance.^{21,22} In contrast, single-crystal cathodes, free from intergranular cracking, offer a more crack-resistant architecture that mitigates these issues and enables clearer observation of chemically driven interfacial effects.²³ Furthermore, oxide-based electrolytes must undergo high-temperature annealing, which often densifies the interface and produces undesirable secondary phases at the cathode.^{24,25} By contrast, sulfide solid electrolytes can be processed at low temperatures, enabling a conformal and intimate contact with the cathode particles. Their relatively compliant mechanical nature minimizes artificial fracture or densification effects, allowing more accurate observation of chemically induced interfacial and morphological changes.²⁶ We employed lithium difluorophosphate (LiDFP, LiPO₂F₂) as a model coating material to suppress chemical degradation²⁷, analyzing reaction uniformity and microstructural changes in composite electrodes with and without the coating.

Added references

21. Cha, H. *et al.* Boosting Reaction Homogeneity in High-Energy Lithium-Ion Battery Cathode Materials. *Adv. Mater.* **32**, 2003040 (2020).
22. Lou, S. *et al.* Insights into interfacial effect and local lithium-ion transport in polycrystalline cathodes of solid-state batteries. *Nat. Commun.* **11**, 1–10 (2020).
23. Qian, G. *et al.* Single-crystal nickel-rich layered-oxide battery cathode materials: synthesis, electrochemistry, and intra-granular fracture. *Energy Storage Mater.* **27**, 140–149 (2020).

Comments:

3. *The concept of studying interface behavior by coating ion conductors on the cathode surface is not novel. The authors need to articulate the core innovation of their work.*

Author reply:

We sincerely thank the reviewer for this important and insightful comment. We fully acknowledge that the application of ionically conductive and electronically insulating coatings—particularly oxide-based materials such as LiNbO₃, Li₂ZrO₃, and Li₄Ti₅O₁₂ (LTO)—has been extensively studied as a means of stabilizing the cathode–electrolyte interface in all-solid-state battery (ASSB) systems. These materials are well known for their structural robustness and capacity to suppress parasitic interfacial reactions. However, oxide-based coatings still suffer from several critical limitations. First, their synthesis typically requires high-temperature solid-state reactions (300–500 °C), which complicates compositional control and often results in the formation of impurity phases at the interface, making it difficult to preserve the designed material properties predicted by theoretical calculations [F. Walther *et al.*, *Chem. Mater.* 2021, 33, 2110–2125; C. Park *et al.*, *Adv. Energy Mater.* 2023, 13, 2203861]. Second, despite structural stability, chemical degradation processes such as cation diffusion from the cathode (e.g., Co migration in LiCoO₂) into the solid electrolyte are not fully

suppressed, leading to progressive performance decay [J. Haruyama *et. al.*, *ACS Appl. Mater. Interfaces* 2017, 9, 286-292]. Third, many oxide coatings itself are electrochemically unstable above 4.0 V, exhibiting oxidative decomposition that distorts the true interfacial behavior and reduces their effectiveness as passivation layers, as demonstrated in **Fig. R5** [A. Banerjee *et. al.*, *ACS Appl. Mater. Interfaces.* 2019, 11, 43138–43145], which can obscure or interfere interfacial insulating effect of coating materials. Finally, the high stiffness of inorganic oxide coatings can lead to crack formation under the anisotropic volume changes of the cathode, thus promoting mechanical failure at the interface and complicating the understanding of intertwined chemo-mechanical degradation [A. Banerjee, *et. al.*, *Chem. Rev.* 2020, 120 6878-6933; W. D. Richards *et. al.*, *Chem. Mater.* 2016, 28, 266-273].

Fig. R5. Necessary criteria for an effective cathode coating for ASSBs.

Referred by [A. Banerjee *et. al.*, *ACS Appl. Mater. Interfaces.* 2019, 11, 43138–43145]

To address these challenges, our group previously introduced organic coating layer such as lithium difluorobis(oxalato)phosphate (LiDFBOP) [C. Park *et. al.*, *Adv. Energy Mater.* 2023, 13, 2203861]. While this approach significantly reduced interface damage during synthesis, LiDFBOP coatings still exhibited partial electrochemical oxidation reaction above 4.1 V (vs. Li⁺/Li), limiting the construction of an intact and uniformly stable interfacial layer under high-voltage conditions. Recently, L. Nazar's group reported the use of lithium difluorophosphate (LiPO₂F₂, LiDFP) as a coating precursor, highlighting its thermodynamic stability over 2.6–4.9 V (vs. Li⁺/Li), which covers the operational range of Ni-rich cathodes [L. Qian *et. al.*, *Angew. Chem. Int. Ed.* 2025, 64, e202413591; see **Fig. R6**]. However, like with oxide coating layer, during the wet synthesis itself, decomposition products such as LiF, Li_xP_yF_x, and LiP_xO_yF_z were generated even before electrochemical cycling commenced, introducing structural and chemical complexities that complicated the interpretation of interfacial behaviors.

The presence of these decomposition species at the interface introduces structural and chemical complexity, making it difficult to interpret chemo-mechanical degradation pathways in a reliable and isolated manner. In contrast, the mechano-fusion method employed in the present study allows the formation of a chemically and electrochemically intact LiDFP interfacial layer, free from premature decomposition and maintaining its stability up to 4.5 V (vs. Li^+/Li), as evidenced in **Fig. R7**. This high chemical fidelity provides a clean model to accurately study reaction dynamics and morphological evolution in ASSBs, overcoming both the oxidation instability of prior organic coatings and the synthesis-induced degradation of coatings process. **Thus, the novelty of our approach lies not merely in the use of a new coating material, but in achieving complete chemical and electrochemical stabilization through controlled synthesis method and material, enabling fundamental insights into chemo-mechanical coupling at solid-state battery interfaces.** This framework allows us to clearly elucidate how interfacial chemistry governs both reaction dynamics and morphological evolution of cathode particles in ASSBs—mechanisms that are often obscured in traditional coating studies due to instability or interfacial mixing. Furthermore, by establishing this well-defined and chemically stable interfacial model, we are able to systematically investigate mechanical pore generation at the cathode surface, free from the confounding effects of chemical decomposition of the surface modification layers.

Fig. R6. Theoretical investigation of LiDFP as a solid-state coating using DFT calculations. a) Electrochemical stability range of LiDFP.

Referred by [L. Qian *et. al.*, *Angew. Chem. Int. Ed.* 2025, 64, e202413591]

Fig. R7. a) Voltage profiles during the initial cycle of NCM composite|LPSCl|In/InLi cells and corresponding b) F 1s XPS spectra of cathode composite after initial cycling of NCM composite|LPSCl|In/InLi full cells, within the voltage range of 1.88–3.88 V (vs. In/InLi) at C/10 (1C = 1.36 mA/cm²) and 25 °C under a stack pressure of 13 MPa.

We really thank the reviewer's comment regarding the clarity of manuscript, and we revised the manuscript as follows:

Original text (Page 5, line 12)

We employed lithium difluorophosphate (LiDFP, LiPO_2F_2) as a model coating material to suppress chemical degradation²¹, analyzing reaction uniformity and microstructural changes in composite electrodes with and without the coating.

Original text (Page 5, line 26)

We employed lithium difluorophosphate (LiDFP, LiPO_2F_2) as a model coating material to suppress chemical degradation²⁷. Unlike conventional oxide-based coatings such as LiNbO_3 or $\text{Li}_4\text{Ti}_5\text{O}_{12}$, which often undergo oxidative decomposition or structural changes under high-voltage and high-pressure conditions^{20,28,29}, LiDFP forms a chemically and electrochemically stable interfacial layer that remains intact even during the cycling at elevated voltages. This stability enables more reliable analysis of reaction uniformity and microstructural changes in composite electrodes with and without the coating.

Added references

28. Banerjee, A. *et al.* Revealing Nanoscale Solid-Solid Interfacial Phenomena for Long-Life and High-Energy All-Solid-State Batteries. *ACS Appl. Mater. Interfaces* **11**, 43138–43145 (2019).

29. Richards, W. D. *et al.* Interface Stability in Solid-State Batteries. *Chem. Mater.* **28**, 266–273 (2016).

30. Qian, L. *et al.* Engineering Stable Decomposition Products on Cathode Surfaces to

Comments:

4. For interface degradation, could more intuitive characterization methods, such as transmission electron microscopy (TEM), be employed?

Author reply:

We sincerely thank the reviewer for the insightful suggestion regarding the use of more intuitive and direct characterization techniques, such as transmission electron microscopy (TEM), to evaluate interfacial degradation. In response, we clarify that we employed aberration-corrected scanning transmission electron microscopy (STEM) coupled with electron energy-loss spectroscopy (EELS) to directly probe both the structural and chemical integrity of the cathode–electrolyte interface after the initial formation cycling. For the bare NCM sample, STEM imaging revealed notable surface degradation, characterized by a phase transformation from the original layered structure to a rock-salt-like structure (**Fig. R8a**). This transformation is widely associated with oxygen loss and interfacial reactions between Ni-rich layered oxide cathodes and sulfide-based solid electrolytes. Such degradation mechanisms have been reported in several previous studies [S. Deng, *et al.*, *Energy Storage Mater.* 2021, 35, 661; E. Gil-González, *et al.*, *Energy Storage Mater.* 2022, 45, 484.].

Further evidence for interfacial degradation was obtained through O–K edge EELS analysis along a surface-to-bulk profile. In the bare NCM, the O–K edge exhibited a distinct rightward shift (toward higher energy loss) at the particle surface compared to the interior (**Fig. R8b**). This shift, particularly in the pre-edge peak (~528.5 eV), reflects modifications in the local oxygen coordination environment, typically attributed to oxygen vacancy formation or partial oxygen depletion. The magnitude of this shift (~3.0 eV) is consistent with a reduction in the average transition metal oxidation state and suggests substantial structural and chemical deterioration from surface to bulk (with about 20 nm thickness). In contrast, for the LiDFP-coated NCM, STEM imaging showed that the layered structure was well preserved after formation cycling, with no evidence of phase transitions or cation disorder (**Fig. R8c**). Corresponding O–K edge EELS spectra from the surface and bulk regions were nearly identical, with a minimal peak shift of less than 1.0 eV (**Fig. R8d**). This uniformity in spectral features implies that the oxygen bonding environment and local electronic structure were effectively retained throughout the particle. These results confirm that the LiDFP coating functions as a chemically and electrochemically stable interfacial layer, suppressing oxygen loss, inhibiting structural transitions, and mitigating degradation reactions at the cathode–electrolyte boundary. Altogether, our combined STEM and EELS analyses provide direct nanoscale evidence for the enhanced interfacial stability of the LiDFP-coated cathode compared to the uncoated system.

Fig. R8. STEM images of NCM cathodes cycled with a) bare NCM and b) LiDFP NCM after initial cycle at C/10 and 25 °C. EELS O K measured from the surface (0 nm) to the inner bulk (20 nm), with a spacing of 2 nm between the measurement points, for c) bare NCM and d) LiDFP NCM after initial cycling.

Comments:

5. Is the interface change during the reaction process a result of the cathode, or is it influenced by factors such as electrolyte decomposition?

Author reply:

We sincerely thank the reviewer for raising this fundamental and highly insightful question. We agree that understanding the origin of interface evolution is crucial for unraveling degradation mechanisms in all-solid-state battery (ASSB) systems. The chemical and mechanical degradation observed at the cathode–electrolyte interface in all-solid-state batteries (ASSBs) arises from a combination of two components, cathode materials and solid electrolytes.

First, on the chemical degradation side, prior studies have demonstrated that argyrodite-type solid electrolytes (e.g., $\text{Li}_6\text{PS}_5\text{Cl}$) undergo decomposition via a thermodynamically favorable indirect pathway, especially under high-voltage or oxidizing conditions [T. K. Schwietert *et al.*, *Nat. Mater.* 2020, 19, 428-435]. This indirect decomposition route has been widely acknowledged as a key contributor to interfacial instability in sulfide-based ASSBs. Building upon this mechanistic insight, we further considered the chemical reactivity of these decomposition byproducts—especially Li_3PS_4 —with the oxide cathode (LiNiO_2). Using thermodynamic modeling informed by prior databases [A. Jain *et al.*, *APL Mater.* 2013, 1,

011002], we evaluated a representative interfacial reaction scenario in which these sulfide and phosphate species subsequently interact with the charged NCM cathode. This leads to the formation of Ni- and S-containing interphases and associated volume contraction. The proposed net reaction is:

This equation highlights that interfacial decomposition is a cooperative process involving degradation of both cathode materials and chemically unstable solid electrolyte components. In this equation, the Ni-rich cathode materials serves as a strong oxidizer, initiating indirect oxidation decomposition of the sulfide electrolyte, resulting in a range of byproducts such as Li_2S , Li_3PO_4 , and Ni_3S_2 , along with elemental sulfur which accompanies degradation of both cathode and SE.

To experimentally validate this mechanism, we performed ToF-SIMS depth profiling. In the bare NCM composite, we observed pronounced surface localization of Li_2S fragments (denoted by LiS^- anion on the spectra), enriched specifically at the cathode–electrolyte interface (**Fig. R9a**). This provides strong evidence that electrolyte decomposition is spatially confirmed at the interfaces between cathode and solid electrolyte. Furthermore, STEM imaging confirmed the development of a rock-salt-like surface structure, consistent with layered-to-rock-salt transformation (**Fig. R9b**). Additionally, aberration-corrected STEM-EELS analysis (focusing on the O–K edge) revealed significant structural and electronic degradation near the surface of bare NCM. The EELS spectra exhibited a distinct shift to higher energy (i.e., right shift) at the surface compared to the bulk region (**Fig. R9c**), which is a well-known signature of oxygen vacancy formation and lattice oxygen loss. These observations support the conclusion that the NCM cathode not only facilitates indirect decomposition of solid electrolyte, but also undergoes its own structural deterioration as a result of interfacial reactions.

Fig. R9. a) TOF-SIMS 3D depth spectra of bare NCM for S^- , LiS^- , PS_3^- , and NiO_2^- anion b)STEM images of NCM cathodes cycled with a) bare NCM after initial cycle at C/10 and 25 °C and c) EELS O K measured from the surface (0 nm) to the inner bulk (20 nm), with a spacing of 2 nm between the measurement points, for bare NCM.

In parallel, mechanical degradation is strongly influenced by both cathode volume change and chemical decomposition of solid electrolyte. For the 4 μm -sized NCM particles used in our study, the reported volumetric expansion during delithiation is approximately 2–6% up to ~80% SOC [R. Koerver *et al.*, *Energy Environ. Sci.* 2018, 11, 2142–2151]. Assuming a spherical particle with a 4 μm diameter, a 2–6% volume change corresponds to a radial expansion of shrinkage of approximately 13.4 nm to 40.8 nm in one direction length. Alongside this effect, the solid electrolyte also undergoes volume change during decomposition. Specifically, the interfacial decomposition reaction described above leads to a net volume contraction of approximately –18.45%. Furthermore, recent studies have reported that $\text{Li}_6\text{PS}_5\text{Cl}$ alone can exhibit a volume reduction as large as –30% upon decomposition, even in the absence of cathode interaction [K.H. Kim *et al.*, *Chem. Mater.* 2024, 36, 5215]. For a representative **1 μm -sized electrolyte particle**, which is representatively used for making a composite with cathode, the values of volume change (%) corresponds to a linear shrinkage of approximately **32.9 nm to 56.1 nm**, indicating significant changes similar to cathode. These estimates underscore that mechanical degradation is not solely driven by cathode volume change but is also substantially affected by electrolyte decomposition. Thus, both electrode and electrolyte must be considered in evaluating contact loss and interfacial instability in ASSB systems.

In conclusion, we emphasize that both chemical and mechanical degradation are governed by both the cathode and the solid electrolyte. Their coupled interaction leads to chemo-mechanical heterogeneity and degradation. This integrated perspective has been reflected in the revised manuscript, with added discussion to clarify this dual contribution.

We really thank the reviewer's comment regarding the clarity of manuscript, and we revised the manuscript as follows:

Original text (Page 20, line 15)

Our statistical analysis of mechanical degradation further reveals that LiDFP NCM exhibits numerous small, uniformly distributed pores around the cathode surface, while bare NCM shows higher variability in pore distribution, accompanied by reaction heterogeneity among particles. This comprehensive micro-scale analysis highlights the need for a refined understanding of the interplay between the chemical and mechanical degradation in ASSBs to effectively comprehend the degradation processes. Additionally, these findings also suggest a crucial role of the coating layer in maintaining lithium-ion diffusion to the cathode while preserving the interface contact, despite particle volume changes.

Revised text (Page 21, line 23)

Our statistical analysis of mechanical degradation further reveals that LiDFP NCM exhibits numerous small, uniformly distributed pores around the cathode surface, while bare NCM shows higher variability in pore distribution, accompanied by reaction heterogeneity among particles. This comprehensive micro-scale analysis highlights the need for a refined understanding of the interplay between the chemical and mechanical degradation in ASSBs to effectively comprehend the degradation processes. **Both chemical and mechanical degradation at the cathode–electrolyte interface are driven by interrelated processes involving the cathode and the solid electrolyte, necessitating a coupled chemo-mechanical interpretation to fully understand interface degradation mechanism in ASSBs.** Additionally,

these findings also suggest a crucial role of the coating layer in maintaining lithium-ion diffusion to the cathode while preserving the interface contact, despite particle volume changes.

Reviewer #2 (Remarks to the Author):

Comments:

In this paper titled "Interfacial Chemistry-Driven Reaction Dynamics and Resultant Microstructural Evolution in All-Solid-State Batteries," the authors examine the complex chemo-mechanical degradation occurring in composite cathodes of sulfide-based ASSBs. To visualize the evolution of the mechanical properties of composite cathodes, they propose a novel and intelligent digital twin 3D reconstruction model based on FIB-SEM images for quantitative microstructure analysis. This approach paves the way for revealing the mechanical properties of the solid-solid contact in ASSBs. However, the analysis of the complex interplay of chemical and mechanical behaviors at the cathode/solid-electrolyte interface is not rigorous. Adjustments or supplements are required to support the hypothesis of this manuscript. This article can be considered for publication in Nature Communications after major revisions. Here are the issues you may refer to:

Author reply:

We are grateful for the valuable and constructive comments provided by the reviewer, especially, feel appreciation for the positive assessment of message that we want to deliver through this study. We tried to do our best with endeavor to address the reviewer's concern/comments as point-by-point responses. In the revision, we conducted additional experiments that can support and reinforce our conclusions. We sincerely hope that this revision relieves the reviewer's concerns.

Comments:

1. The authors employed the mechano-fusion method to synthesize the LiDFP coating NCM cathode. Supplementary Fig. 1a confirmed a homogeneous and thin LiDFP-based film (approximately 10 nm thick) on the cathode surface. However, the morphological properties of the LiDFP and LiDFP NCM on the micrometer scale should also be provided so that the changes can be simultaneously comprehended after mechanical mixing.

Author reply:

We sincerely thank the reviewer for this constructive suggestion. To address this point, we have now included comparative morphological data of LiDFP powder, bare NCM, and LiDFP-coated NCM as shown in **Fig. R10a-c** (which denoted as Supplementary Fig. 1 in revised supplementary information). As shown, the pristine LiDFP powder exhibits a distinct and separate particle morphology. However, after the mechano-fusion coating process, no discernible LiDFP particles are visible on the NCM surface, indicating successful and conformal dispersion of the coating material. Additionally, we note that the SEM images of the LiDFP-coated NCM cathode display slightly blurred image compared to the bare NCM. This phenomenon is frequently observed in SEM imaging of materials coated with electronically insulating. This further supports the successful and conformal deposition of electronic insulating LiDFP coating on the cathode surface (**Fig. R10c**).

Furthermore, as shown in **Fig. R10d**, EDS elemental mapping clearly confirms the homogeneous distribution of P and F elements—originating from LiDFP coating material—on the NCM surface, validating the formation of a uniform LiDFP-based surface layer.

Importantly, SEM images show that the micrometer-scale morphology and particle size of NCM remain intact after the coating process. This suggests that the mechano-fusion approach does not induce particle agglomeration or structural distortion at the microscale particle level.

Fig. R10. Top-view SEM images of a) Bare NCM, b) LiDFP NCM and c) LiDFP coating material. d) EDS mappings of LiDFP NCM, showing the distribution of Ni, O, F, P elements.

These findings collectively demonstrate that the coating is both morphologically conformal and chemically uniform without affecting the bulk morphology of the NCM cathode. We have updated the Supplementary Information and corresponding figure captions to clarify these results as follow:

Original text (Page 6, line 25)

We employed the mechano-fusion method, which utilizes shear force friction between cathode and coating materials to create a uniform interfacial layer without morphological or compositional changes.²⁰ TEM analysis confirmed the presence of a homogeneous and thin LiDFP-based film (approximately 10 nm thick) on the cathode surface (Supplementary Fig. 1a).

Revised text (Page 7, line 16)

We employed the mechano-fusion method, which utilizes shear force friction between cathode and coating materials to create a uniform interfacial layer without morphological or compositional changes (Supplementary Fig. 1).²⁰ TEM analysis confirmed the presence of a

homogeneous and thin LiDFP-based film (approximately 10 nm thick) on the cathode surface (Supplementary Fig. 2a).

Added Figure (Supplementary Fig. 1, Revised Supplementary Information, page 3)

Supplementary Fig. 1. Top-view SEM images of a) Bare NCM, b) LiDFP NCM and c) LiDFP coating material. d) EDS mappings of LiDFP NCM, showing the distribution of Ni, O, F, P elements.

Comments:

2. The electrochemical properties shown in Figures 1. e and 1. f are inconsistent. Reproducible raw data should be provided to clarify the properties' differences between the modified and pristine samples, ensuring self-consistency within this manuscript. The same concern is also present in Figure 4. i.

Author reply:

We sincerely thank the reviewer for the careful observation and critical comment regarding the consistency of electrochemical performance data. We would like to clarify that the apparent difference in discharge capacity at the same C/2 rate between Fig. 1e (rate test) and Fig. 1f (long-term cycling) originates from the fact that both measurements were conducted sequentially on the same cell, not independently on separate cells.

Specifically, the long-term cycling shown in Fig. 1f was initiated immediately after the rate test as shown in Fig. 1e. As a result, the slightly lower initial capacity at C/2 in Fig. 1f compared to Fig. 1e is attributed to irreversible capacity loss incurred during the prior high-rate testing. This is a common phenomenon in solid-state batteries, where prior electrochemical history can

affect subsequent electrochemical performance. To validate the reproducibility of our findings, we conducted additional electrochemical measurements on independently assembled cells for both bare NCM and LiDFP-coated NCM, following identical protocols. The results are presented in **Fig. R11** and **Fig. R12**, demonstrating minimal variation in discharge capacity across replicates. Notably, the reproducible data for bare NCM show a maximum capacity deviation of approximately 15 mAh g^{-1} at the same C-rate. This is well within the variation range recently reported by Janek's group [M. Kissel *et al.* *Adv. Energy Mater.* 2025, 2405405], where solid-state cells fabricated *via* manual (hand) mixing exhibited capacity variations up to 28 mAh g^{-1} , primarily due to local inhomogeneities in particle contact and dispersion. Therefore, the consistency of our replicate data confirms that the original figures reflect representative behavior, and the small variations are not anomalous but within acceptable reproducibility standards for ASSBs.

Fig. R11. a) Reproducible electrochemical cycle performance of bare NCM. b) Rate capability of bare NCM at varying C-rates: C/10, C/4, C/2, 1C, 2C, 3C, and back to C/10 and substantial c) long-term cycle performance at C/2 between bare NCM at 25 °C under a stack pressure of 120 MPa with two reproducible cells.

Fig. R12. a) Reproducible electrochemical cycle performance of LiDFP NCM. b) Rate capability of LiDFP NCM at varying C-rates: C/10, C/4, C/2, 1C, 2C, 3C, and back to C/10 and substantial c) long-term cycle performance at C/2 between bare NCM at 25 °C under a stack pressure of 120 MPa with two reproducible cells.

Additionally, to address the reviewer's concern regarding the Fig. 4i, we have performed further reproducibility test under low stack pressure conditions using multiple independently assembled cells. As shown in **Fig. R13**, the capacity trends under low stack pressure were consistent across all replicates, with negligible cell-to-cell variation. This confirms that the

performance differences observed in our study are intrinsic and not due to experimental inconsistency.

Fig. R13. Reproducible electrochemical cycle performance of bare NCM and LiDFP NCM at C/10 ($1C = 1.36 \text{ mA/cm}^2$) rate and $25 \text{ }^\circ\text{C}$ under varying stack pressure of a) 2.6, b) 3.3, and c) 6.5 MPa.

We really thank the reviewer's comment regarding the clarity of manuscript, and we revised the manuscript as follows:

Original Figure caption (page 34)

Fig. 1. Chemical degradation mitigation through surface modification and corresponding electrochemical properties. a) Current–time curves of the cathode-SE composite (65:35 wt%) under a polarization of 200 mV in a steel|NCM composite|steel ion-blocking cell, and b) current–time curves of the cathode-SE composite (65:35 wt%) under a polarization of 50 mV in an electron-blocking cell, composed of In/InLi|LPSCI|NCM composite|LPSCI|In/InLi, for both bare NCM and LiDFP NCM. TOF-SIMS 3D depth spectra and 3D images of c) bare NCM and d) LiDFP NCM for S^- , LiS^- , PS_3^- , and NiO_2^- anion after the first cycling. Comparison of e) rate capability of bare NCM and LiDFP NCM at varying C-rates: C/10, C/4, C/2, 1C, 2C, 3C, and back to C/10. f) Long-term cycle performance at C/2 between bare NCM and LiDFP NCM at $25 \text{ }^\circ\text{C}$ under a stack pressure of 120 MPa.

Revised Figure caption (page 38)

Fig. 1. Chemical degradation mitigation through surface modification and corresponding electrochemical properties. a) Current–time curves of the cathode-SE composite (65:35 wt%) under a polarization of 200 mV in a steel|NCM composite|steel ion-blocking cell, and b) current–time curves of the cathode-SE composite (65:35 wt%) under a polarization of 50 mV in an electron-blocking cell, composed of In/InLi|LPSCI|NCM composite|LPSCI|In/InLi, for both bare NCM and LiDFP NCM. TOF-SIMS 3D depth spectra and 3D images of c) bare NCM and d) LiDFP NCM for S^- , LiS^- , PS_3^- , and NiO_2^- anion after the first cycling. Comparison of e) rate capability of bare NCM and LiDFP NCM at varying C-rates: C/10, C/4, C/2, 1C, 2C, 3C, and back to C/10 and subsequent f) long-

term cycle performance at C/2 between bare NCM and LiDFP NCM at 25 °C under a stack pressure of 120 MPa with 7.4 mg/cm² active material mass loading.

Comments:

3. The basis for Equation 5 in the manuscript concerning the chemical reactions of the Li₆PS₅Cl electrolyte and cathode and the volume change ratio needs to be proven or cited. Additionally, there is a minor clerical error in the writing of Equation 5. Please review the entire manuscript for clarity and accuracy!

Author reply:

We sincerely thank the reviewer for the careful assessment of Equation 5 and the valuable suggestion to clarify the origin and validity of the chemical reaction and associated volume change. We have revised the manuscript accordingly and provide a detailed justification below.

The reaction presented in Equation 5 is based on a stepwise degradation mechanism involving both the electrochemical decomposition of the sulfide electrolyte (Li₆PS₅Cl) and its subsequent chemical reaction with the LiNiO₂ cathode material as the representative for high Ni-based cathode. The sulfide solid electrolytes, Li₆PS₅Cl, is known to decompose ultimately yielding Li₃PS₄, LiCl, and S as final decomposition products as shown in equation (R1) (**Fig. R14a**) [T. K. Schwietert *et. al.*, *Nat. Mater.* 2020, 19, 428-435]. Building upon this mechanistic insight, we further considered the chemical reactivity of these decomposition byproducts especially Li₃PS₄ with the oxide cathode (LiNiO₂). Using thermodynamic modeling informed by prior databases (**Fig. R14b**) [A. Jain *et. al.*, *APL Mater.* 2013, 1, 011002]:

Fig. R14. a) Formation energies per formula unit for all Li configurations within one unit cell versus the composition x in $\text{Li}_x\text{PS}_5\text{Cl}$. Referred by [T. K. Schwietert *et. al.*, *Nat. Mater.* 2020, 19, 428-435]. b) Reaction energy landscape at Li_3PS_4 and LiNiO_2 interface.

These reactions collectively explain the emergence of interfacial degradation products such as Ni_3S_2 , Li_3PO_4 , Li_2SO_4 , and Li_2S shown in equation (R3). As a result, interfacial reaction between Ni-rich cathode and sulfide solid electrolytes ($\text{Li}_6\text{PS}_5\text{Cl}$) accompanied by the volume change (of 18.45 %) at point contact with cathode materials and sulfide solid electrolytes.

This volumetric contraction is primarily driven by the formation of denser products such as Ni_3S_2 and Li_3PO_4 at the interface and our calculation aligns with prior studies [K.H. Kim *et al.*, *Chem. Mater.* 2024, 36, 5215], which reported that the decomposition of $\text{Li}_6\text{PS}_5\text{Cl}$ alone leads to a substantial volume reduction (about -30%) during interfacial degradation. Thus, the ~18% volume decrease obtained from theoretical calculation under the suggested reaction mechanism is reliable.

Fig. R15. Schematic Illustration of the Interfacial Degradation Modes of LPSCl at the Cathode Surface in LPSCl-Based ASSBs. . Referred by [K.H. Kim *et al.*, *Chem. Mater.* 2024, 36, 5215].

We really thank the reviewer's comment regarding the clarity of manuscript, and we revised the manuscript as follows:

Original text (Page 12, line 3)

In addition, chemical-reaction-induced mechanical degradation³², resulting from volume reduction (as described by reaction equation (5)) exacerbates charge-transfer kinetics deviation among cathode particles. Random point contacts between CAM and SE further contribute to heterogeneous charge-transfer kinetics.

Revised text (Page 12, line 23)

In addition, chemical-reaction-induced mechanical degradation⁴¹, resulting from volume reduction **with consecutive reactions** (as described by reaction equation (5) - (7)) exacerbates charge-transfer kinetics deviation among cathode particles.⁴²⁻⁴⁴ Random point contacts between CAM and SE further contribute to heterogeneous charge-transfer kinetics.

$$(\Delta V = -18.45\%) \quad (7)$$

Added references

42. Schwietert, T. K. *et al.* Clarifying the relationship between redox activity and electrochemical stability in solid electrolytes. *Nat. Mater.* **19**, 428–435 (2020).

43. Kim, K. *et al.* Interfacial Degradation Mechanism of Nanostructured LiCoO₂ for Li₆PS₅Cl-Based All-Solid-State Batteries. *Chem. Mater.* **36**, 5215–5227 (2024).

44. Jain, A. *et al.* Commentary: The materials project: A materials genome approach to accelerating materials innovation. *APL Mater.* **1**, 011002 (2013).

Comments:

4. The reasons for the leaning toward a lower angle of Ni K-edge X-ray absorption near edge structure spectrum (XANES) at 0% SOC after LiDFP coated should be valued. This phenomenon may be caused by the valance distribution of nickel ions, the evolution of the surface structure of the NCM cathode materials, and other contributing factors. Additionally, it is essential to consider the effect of the electronegativity of the ligand surrounding Ni and the coordination structure. This aspect should be included in the elaboration. If not addressed, this could result in a logical disconnect.

Author reply:

We sincerely thank the reviewer for this important and insightful comment regarding the interpretation of the Ni K-edge XANES spectra and the origin of the observed edge shift.

First, we apologize for the confusion caused by the label “0% SOC” in Fig. 2. As clarified in the revised version, this point refers not to the pristine (uncycled) state, but to the end-of-discharge state after the first cycle. We have accordingly updated the figure label and manuscript text to describe this more accurately as “end-of-discharge after the first cycle” (see **Fig. R16**). Based on the coulombic efficiency (75.1% for bare NCM and 79.9% for LiDFP-coated NCM), the corresponding SOC values are approximately 24.9% and 20.1%, respectively.

Fig. R16. Voltage profile of NCM composite|LPSCI|In/InLi with bare NCM and LiDFP NCM within the voltage range of 1.88–3.88 V (vs. In/InLi) under a current density of 0.48 mA/cm² at 25 °C with a stack pressure of 13 MPa for a) voltage vs. time and b) capacity vs. voltage

Given the higher reversible capacity of the LiDFP-coated cathode, it undergoes more extensive reduction at the end of the first discharge. Consequently, the average oxidation state of Ni in the coated sample is lower than that in the bare sample. This difference accounts for the observed shift of the Ni K-edge XANES spectrum toward lower energy, consistent with a greater proportion of Ni²⁺ species and deeper lithiation.

Beyond redox behavior, we fully agree with the reviewer that changes in the local coordination environment—particularly the electronegativity and bonding characteristics of surrounding ligands—can also influence the XANES spectral position. In our system, the LiDFP coating introduces phosphate and fluorinated species (e.g., PO₄²⁻ and F⁻) at the cathode surface, while in the bare NCM sample, sulfide-derived species such as LiS⁻ may accumulate due to interfacial decomposition of the solid electrolyte. These interfacial changes could potentially perturb the local ligand field surrounding Ni atoms. However, since hard X-ray XANES is predominantly bulk-sensitive, the contribution from near-surface coordination changes is expected to be limited [Bunker, G., Introduction to XAFS, Cambridge University Press (2010); de Groot, F.M.F., Core Level Spectroscopy of Solids, CRC Press (2008)]. Therefore, we attribute the dominant cause of the Ni K-edge shift to the variation in the average oxidation state of Ni in the bulk, which in turn reflects the extent of lithiation and overall electrochemical reversibility influenced by the surface coating.

We are grateful to the reviewer for raising this nuanced and meaningful point, which allowed us to refine our interpretation and improve the clarity of our discussion as follows:

Original text (Page 11, line 21)

The bare NCM showed elevated DOH values exceeding 50 at SOC 40, SOC 80, and SOC 0 (end of discharge). In contrast, LiDFP NCM maintained DOH values below 40 across all SOC ranges, indicating a more uniform lithium distribution throughout the cathode material (Fig. 2d).

Revised text (Page 12, line 16)

The bare NCM showed elevated DOH values exceeding 50 at SOC 40, SOC 80, and end of

discharge after the first cycle (SOC 24.9 for bare NCM SOC 20.1 for LiDFP NCM). In contrast, LiDFP NCM maintained DOH values below 40 across all SOC ranges, indicating a more uniform lithium distribution throughout the cathode material (Fig. 2d).

Original text (Page 13, line 12)

At SOC 0, bare NCM follows a general normal distribution, but with considerable width due to charge heterogeneity and elevated average SOC, reflecting an increased irreversible capacity (Fig. 2i).

Revised text (Page 14, line 12)

At the end of discharge after the first cycle, bare NCM follows a general normal distribution, but with considerable width due to charge heterogeneity and elevated average SOC, reflecting an increased irreversible capacity (Fig. 2i).

Original Figure (page 35)

Original Caption (page 35)

Fig. 2. Cathode reaction heterogeneity as influenced by chemical degradation. a) Voltage profile of NCM composite|LPSCI|In/InLi with bare NCM and LiDfP NCM within the voltage range of 1.88–3.88 V (vs. In/InLi) under a current density of 0.48 mA/cm² at 25 °C with a stack pressure of 13 MPa, along with a corresponding contour plot depicting the diffraction peak evolution of the (003) planes for b) bare NCM and c) LiDfP NCM during *in-situ* XRD measurement. d) Corresponding DOH parameter acquired from *in-situ* XRD measurement. TXM-XANES mapping at low magnification with multiple cathode particles at SOC 40, SOC 80, and end of discharge (SOC 0) for e) bare NCM and f) LiDfP NCM with mean maximum energy value and standard deviation. Corresponding normalized energy value histogram of pixel from TXM image in g) SOC 40, h) SOC 80, and i) end of discharge (SOC 0).

Revised Figure (page 39)

Revised Caption (page 39)

Fig. 2. Cathode reaction heterogeneity as influenced by chemical degradation. a) Voltage profile of NCM composite|LPSCl|In/InLi with bare NCM and LiDFP NCM within the voltage range of 1.88–3.88 V (vs. In/InLi) under a current density of 0.48 mA/cm² at 25 °C with a stack pressure of 13 MPa, along with a corresponding contour plot depicting the diffraction peak evolution of the (003) planes for b) bare NCM and c) LiDFP NCM during *in-situ* XRD measurement. d) Corresponding DOH parameter acquired from *in-situ* XRD measurement. TXM-XANES mapping at low magnification with multiple cathode particles at SOC 40, SOC 80, and end of discharge after the first cycle for e) bare NCM and f) LiDFP NCM with mean maximum energy value and standard deviation. Corresponding normalized energy value histogram of pixel from TXM image in g) SOC 40, h) SOC 80, and i) end of discharge after the first cycle.

Comments:

5. The assumption of "Despite these limitations, we posit that a relative comparison of active surface area between bare and LiDFP NCM, as an assessment of mechanical degradation, is feasible using the following equations" lacks a degree of rigor. The active contact area and the ion diffusion coefficient have previously been interlinked. Therefore, a more thorough explanation may be warranted.

Author reply:

We sincerely thank the reviewer for this insightful and rigorous comment. We fully acknowledge the importance of critically evaluating the assumptions underlying our interpretation of GITT-derived parameters—particularly in regard to the complex interplay between active surface area and lithium-ion diffusivity, which is a central issue in all-solid-state battery (ASSB) systems where interface limitations often dominate overall kinetics.

To clarify, the lithium-ion diffusion coefficient (D) extracted from GITT fundamentally reflects bulk transport properties, which are primarily governed by the lithium content (i.e., state of charge, SOC) within the active material. In our analysis, all GITT measurements were conducted under tightly controlled SOC conditions between bare and LiDFP-coated NCM samples. As shown in **Fig. R17**, a duration time of 50 min ensures the relaxation of cathode particles in designated SOC. This ensures that bulk lithium content and chemical potential are effectively matched, thereby minimizing any intrinsic variation in D due to state-dependent diffusivity.

Given that the LiDFP coating is a nanometer-thick surface layer, it is unlikely to significantly alter the intrinsic bulk diffusion coefficient. Thus, under constant SOC, any variation in the apparent GITT response (i.e., the slope of voltage relaxation) can be primarily attributed to differences in electrochemically active surface area (S) rather than changes in D itself. In line with previous studies, the relevant GITT expression—commonly rearranged as $D^{\text{GITT}} \times S^2$ —supports the feasibility of “relative” comparison of active surface area when D is assumed to be equivalent across samples at the same SOC (equation R4 and R5).

Fig. R17. Contour plot of the diffraction peak evolution of the (003) planes for fresh bare NCM with 1h rest after charge and discharge process.

$$\left(\frac{\Delta E_s}{\Delta E_t}\right)^2 = k \times D^{GITT} \times S^2 \quad (\text{R5})$$

$$k = \frac{\pi\tau}{4n_m^2 V_M^2}$$

(R6)

To further substantiate this approach, we have reviewed and now referenced several key studies that adopt a similar methodology in ASSB systems. These works apply GITT not only to evaluate bulk diffusivity but also to infer relative changes in electrochemically active surface area or contact quality at interfaces (see Table R1).

System Description	Active Surface ratio	Cell performance	GITT Analysis Data	Reference

SE-coated LCO vs Bare LCO	81% vs 31%		Adv. Mater. 2016, 28, 1874
LPSCI-infiltrated LCO vs Dry-mixed LCO	61% vs 31%		Nano Lett. 2017, 17, 3013
NCM622 / Coarse / Fine electrolyte ratio (70 / 0 / 28) (70 / 7 / 21) (70 / 14 / 14) (70 / 21 / 7) (70 / 28 / 0) + 2wt% carbon	39.32% 46.46% 46.70% 62.04% 50.03%		J. Electrochem. Soc., 2019, 166 (3) 5318

3.5wt% NBR- LiG3 NCM622 vs Bare NCM622	42% vs 27%			Adv. Energy Mater. 2019, 9, 1802927
Ionomer + NCM712 vs PTFE + NCM712 vs Bare NCM712	2.17S vs 1.52S vs 1.0S			ACS Energy Lett. 2022, 7, 1092
0.1wt% CNT coated LCO vs 0.3wt% CNT coated LCO	1.4S vs 1.0S			Chem. Eng. J. 2025, 511, 162096

We really thank the reviewer's comment regarding the clarity of manuscript, and we revised the manuscript as follows::

Original text (Page 18, line 19)

It is important to note that in ASSB systems, the inherent nature of point contacts results in

incomplete assurance of active surface area, presenting significant challenges in accurately determining lithium diffusivity via GITT.⁵⁸ Despite these limitations, we posit that a relative comparison of active surface area between bare and LiDFP NCM, as an assessment of mechanical degradation, is feasible using the following equations (detailed variable explanations are provided in Supplementary Table S5).

Revised text (Page 19, line 22)

It is important to note that in ASSB systems, the inherent nature of point contacts results in incomplete assurance of active surface area, presenting significant challenges in accurately determining lithium diffusivity via GITT.⁵⁸ Despite these limitations, we posit that a relative comparison of active surface area between bare and LiDFP NCM, as an assessment of mechanical degradation, is feasible using the following equations (detailed variable explanations are provided in Supplementary Table S5). Note that the diffusion coefficient (D) obtained from GITT is an intrinsic bulk property. Because the coating is only a few nanometres thick and leaves the lithium-ion pathways in the bulk unchanged, the diffusivity should be identical for samples compared at the same state of charge (SOC). Consequently, variations in the voltage-relaxation profiles mainly indicate differences in the electrochemical contact area (S), rather than the intrinsic lithium diffusivity.^{58–60}

Added references

58. Oh, D. Y. *et al.* Slurry-Fabricable Li⁺-Conductive Polymeric Binders for Practical All-Solid-State Lithium-Ion Batteries Enabled by Solvate Ionic Liquids. *Adv. Energy Mater.* **9**, 1802927 (2019).

59. Hong, S. B. *et al.* All-Solid-State Lithium Batteries: Li⁺-Conducting Ionomer Binder for Dry-Processed Composite Cathodes. *ACS Energy Lett.* **7**, 1092–1100 (2022).

60. Park, K. H. *et al.* Solution-Processable Glass LiI-Li₄SnS₄ Superionic Conductors for All-Solid-State Li-Ion Batteries. *Adv. Mater.* **28**, 1874–1883 (2016).

Comments:

6. The enhanced performance of LiDFP NCMs primarily stems from their capacity to maintain active contact surfaces and facilitate lithium-ion diffusion. Is it possible to effectively quantify how the active surface contributes to the performance of anode materials?

Author reply:

We sincerely thank the reviewer for raising this insightful question. The contribution of the active surface area to electrochemical performance is particularly important in all-solid-state battery systems, where the interface between the cathode and solid electrolyte is often limited and discontinuous due to the absence of liquid-phase infiltration.

In our study, we estimated the relative difference in active contact area between the coated and uncoated cathode electrodes by analyzing the squared ratio of equilibrium to transient voltage changes obtained from the galvanostatic intermittent titration (GITT) technique. The coated sample showed approximately 20.2 % higher value, which implies improved interfacial contact after the application of the coating layer based on electrochemical test.

However, this difference in active surface area does not linearly translate into proportional enhancement in discharge capacity. For instance, the relative improvement in capacity observed in the coated electrode—14.17 % at the lowest current rate of C/10 and up to 178.89% at higher current rates of 1C-rate—suggests that the relationship between active contact area and electrochemical performance is nonlinear. This trend is consistent with earlier reports, including that of Hong and co-workers, who observed that a 117 % increase in contact area did not yield an equivalent increase in performance [S. B. Hong *et al.*, *ACS Energy Lett.* 2022, 7, 1092–1100; D. Y. Oh *et al.* *Adv. Energy Mater.* 2019, 9, 1802927].

This nonlinearity arises because battery performance is governed not only by the available contact area for charge transfer but also by other resistive elements in the system. These include bulk resistance in the solid electrolyte, interfacial charge-transfer resistance that depends on decomposition products at the surfaces, ionic mass transport resistance, and current density. Each of these parameter changes depending on the electrochemical state of the cell and is challenging to fully separate.

To further understand the role of contact area, we prepared two electrode composites with different cathode-to-electrolyte ratios (65:30:5 *vs.* 50:45:5 by weight). While using the same cathode material, the composite with higher electrolyte content exhibited a 26.4 % higher contact-area-related voltage ratio and correspondingly improved rate capability and capacity retention. This demonstrates that the extent of electrochemically active contact is vital factor that demonstrates the electrochemical properties of the cells.

Fig. R18. Comparison of electrochemical cell performance according to composition of cathode composite at 13 MPa stack pressure environment. a) GITT voltage profiles and b) corresponding $(\Delta E_s/\Delta E_t)^2$ values of bare NCM composite with varying cathode-to-electrolyte ratios (65:30:5 *vs.* 50:45:5 wt%). c) Rate performance comparison between cells with two different cathode composite ratios.

Although precise quantification of the contribution from active contact area would require the decoupling of all individual resistance components and hard to define, which remains nontrivial in all-solid-state systems, our experimental results confirm that enhanced interfacial contact contributes meaningfully to the improved electrochemical behavior observed in coated cathodes.

Comments:

7. 3D reconstruction using FIB-SEM effectively facilitates quantitative microstructural analysis. However, the use of e.g. U-net diffusion models for image optimization in this technique may result in variations in microregion pixel points. Do the authors have any comments on the error evaluation of the model?

Author reply:

We sincerely thank the reviewer for this thoughtful and technically relevant comment. We fully agree that evaluating the accuracy and reliability of image segmentation—particularly when using data-driven approaches such as U-net—is essential for ensuring the validity of subsequent microstructural quantification.

As noted in recent literature, three representative approaches are commonly used to assess segmentation accuracy:

- 1) Comparing pixel-wise agreement ratio between the segmentation results and the ground truth. [K. Choudhary *et. al.* *NPJ Computational Mater.*, 2022, 8, 194]
- 2) Comparing the segmented volume fractions to experimentally input volume fractions. [M. Kroll. *et.al.* *J. Power Sources Adv.*, 2021, 505, 230064, M. Kodama. *et. al.* *J. Power Sources Adv.*, 2021, 8, 100048]
- 3) In the field of data science, segmentation accuracy is evaluated using various methods such as the Dice score and Intersection-over-Union (IoU), etc [Z. Su. *et. al.* *npj Computational Mater.*, 2022, 8, 30]

In our study, we adopted the second approach, which is widely employed in the field of battery materials research.

Specifically, as shown in **Fig. R19**, the cathode composite was fabricated using an experimentally controlled mass ratio of NCM622: LPSCI: CNF = 65:30:5 (wt%). This corresponds to a theoretical volume ratio of 39.79: 52.64: 7.57 (vol%), based on their respective densities (NCM622: 4.7 g cm⁻³, LPSCI: 1.64 g cm⁻³, CNF: 1.9 g cm⁻³).

In our reconstructed digital twin models, the volume ratio of NCM622: LPSCI: CNF for LiDFP coated NCM composite was 39.06: 51.30: 9.64 (vol%) and 39.50: 54.85: 5.65 (vol%) for bare NCM composite. Considering the density of each material, the weight ratio of NCM622: LPSCI: CNF was 64.19: 29.41: 6.40 (wt%) for LiDFP coated NCM composite and 64.84: 31.41: 3.75 for bare NCM composite. The volume/weight ratio obtained from the digital twin model closely matches the theoretical volume ratio based on the experimentally used composition (**Fig. R19**).

Fig. R19. Comparison of weight ratio of cathode composite component. [M. Kroll. *et.al. J. Power Sources Adv.*, 2021, 505, 230064; M. Kodama. *et. al. J. Power Sources Adv.*, 2021, 8, 100048; S. Choi. *et. al, ACS Appl. Mater. Interfaces*, 2018, 10, 23740; P. Pinnmann. *et. al. J. Electrochem. Soc.*, 2024, 171, 060514; and T. Li. *et. al. ACS Appl. Mater. Interfaces* 2018, 10, 16927]

These values show excellent agreement (under 6 % discrepancies between segmentation and experimental data) with the target theoretical values derived from experimental inputs, indicating that our segmentation model provides quantitatively reliable representations of the composite microstructure.

We really thank the reviewer’s comment regarding the clarity of manuscript, and we revised the manuscript as follows:

Original text (Page 24, line 10)

For image segmentation, we adapted the convolutional neural network (CNN)-based U-net algorithm, commonly utilized in biomedical image segmentation, for the composite cathodes of ASSBs. Through U-net training, we segmented NCM, LPSCI, CNF, and Pore, constructing a digital twin structure from the results. This digital twin structure was then used for simulation and calculation. The electronic and ionic current density maps were calculated using Ohm's law, with $\Delta V = 1$ V under Dirichlet boundary conditions, and this calculation was implemented using the explicit jump (EJ) solver of the ConductoDict module in GeoDict2024.

Revised text (Page 26, line 12)

For image segmentation, we adapted the convolutional neural network (CNN)-based U-net algorithm, commonly utilized in biomedical image segmentation, for the composite cathodes of ASSBs. Through U-net training, we segmented NCM, LPSCI, CNF, and Pore, constructing a digital twin structure from the results. This digital twin structure was then

used for simulation and calculation. To evaluate the segmentation accuracy of the U-net-based 3D reconstruction, we adopted the commonly used validation method—comparing the segmented volume fractions with experimentally input values based on material densities. These values show excellent agreement, with discrepancies less than 6% between the segmentation-derived and actual experimental volume ratios, confirming the reliability of the digital twin model. The electronic and ionic current density maps were calculated using Ohm's law, with $\Delta V = 1$ V under Dirichlet boundary conditions, and this calculation was implemented using the explicit jump (EJ) solver of the ConductoDict module in GeoDict2024.

Reviewer #3 (Remarks to the Author):

Comments:

All-solid-state batteries based on sulfide electrolytes attained enormous attentions owing to their high energy density and high intrinsic safety. Nevertheless, the complex microstructural evolution including pore formation and contact failure originated from chemical/electrochemical instabilities of cathodes/solid electrolytes interfaces and breath effect of cathodes, hinder the enhancement for battery performance. Herein, the authors investigate the effect of interfacial attenuation on electrochemical properties of cathodes and the mechanism of microstructure evolution, by constructing a well-defined model system (LiDFP@NCM622). The whole work is impressive and valuable. But there are still some problems to be addressed before the consideration for publication.

Author reply:

We appreciate the reviewer for careful review on our manuscript and delivering helpful comments. In the following, we tried our best to address the reviewer's concern and clarify our argument with additional experiment. We hope that the manuscript is now ready to be published in *Nature Communications*. We reproduced the comments of the reviewer, responded to the comments and suggestions.

Comments:

1. The well-defined model has been constructed with uniform LiDFP coating layer onto NCM. There are clear observations in Fig. S1a, but the elemental distributions of cross-sectioned STEM image are still in need for the precise distinction of heterojunction structure.

Author reply:

We sincerely thank the reviewer for this insightful and constructive comment regarding the characterization of the coating layer. We fully agree that cross-sectional STEM-EDS elemental mapping offers a more direct and precise assessment of the heterojunction structure and provides critical validation of the uniformity of the LiDFP coating on NCM particles. In response to the reviewer's suggestion, we performed cross-sectional STEM imaging and EDS elemental mapping, which are now included in the revised Supplementary Fig. 2. As shown in **Fig. R20**, the elemental signals of Ni (from the NCM cathode) and P (originating from the LiDFP coating) are distinctly observed along the particle surfaces, indicating the formation of a uniform and continuous coating layer approximately 5–10 nm thick, with no observable phase transformation or compositional inhomogeneity. In addition, Fast Fourier Transform (FFT) analysis of the STEM images reveals a well-ordered layered diffraction pattern characteristic of the crystalline NCM, alongside ring-shaped amorphous features, consistent with the formation of a conformal and amorphous LiDFP coating at the heterojunction interface (**Fig. R20a**). These results confirm that the LiDFP coating is chemically homogeneous, structurally intact, and uniformly distributed, supporting the presence of a precisely formed heterointerface. This is in excellent agreement with prior surface observations made using TEM and surface-sensitive ToF-SIMS analysis (**Fig. R20**). In particular, the detection of characteristic PO_2^- and PO_2F_2^- anion fragments in the coated samples further confirms the chemical identity and integrity of the LiDFP layer without detectable decomposition or alteration after synthesis.

Fig. R20. a) Cross-sectioned STEM image of NCM surface coated with 0.3 wt% LiDFP, denoted as LiDFP NCM. b) Cross-sectioned STEM image and corresponding EDS mapping for Ni and P atoms. TOF-SIMS surface spectra of LiDFP additive, bare NCM, and LiDFP NCM of b) PO_2^- anion and c) PO_2F_2^- anion.

Taken together, the combination of surface morphology, chemical composition mapping, and reciprocal-space analysis provides compelling evidence for the successful construction of a well-defined, chemically intact LiDFP coating layer at the cathode surface. We have revised the Supplementary Fig. 2 in revised Supplementary Information accordingly to clearly reflect these findings and directly address the reviewer's request as follows:

Original figure (Supplementary Fig. 1., Supplementary Information, page 3)

Supplementary Fig. 1. a) Cross-sectioned STEM image of NCM surface coated with 0.3 wt% LiDFP, denoted as LiDFP NCM. TOF-SIMS surface spectra of LiDFP additive, bare NCM, and LiDFP NCM of b) PO_2^- anion and c) PO_2F_2^- anion.

Revised Figure (Supplementary Fig. 2., Supplementary Information, page 3)

Supplementary Fig. 2. a) Cross-sectioned STEM image and corresponding FFT image of NCM surface coated with 0.3 wt% LiDFP, denoted as LiDFP NCM. b) Cross-sectioned STEM image and corresponding EDS mapping for Ni and P atoms. TOF-SIMS surface spectra of LiDFP additive, bare NCM, and LiDFP NCM of c) PO₂⁻ anion and d) PO₂F₂⁻ anion.

Comments:

2. The detailed assembly processes of all-solid-state batteries for in-situ XRD testing of should be stated. At the same time, in the XRD test, considering that the battery contains a variety of components, the signals representing other components besides cathode actually exist, and how the authors distinguish and exclude them should be indicated. Relevant theoretical analysis and systematic experimental demonstration should be given.

Author reply:

We sincerely thank the reviewer for the valuable comment regarding the configuration and interpretation of our in-situ XRD measurements. We fully recognize the reviewer's concern that various components in the all-solid-state battery (ASSB)—such as the electrolyte, counter electrode (in this case for In/InLi alloy), and current collector—could potentially contribute to the overall diffraction signal, complicating the attribution of structural changes to the cathode material alone. To address this concern, we carefully designed the cell architecture for in-situ analysis. The pellet was assembled in the following sequence: a Li–In alloy foil (counter electrode), a solid electrolyte layer (~200 mg), a cathode composite layer (~20 mg), and finally an Al foil (current collector). This stack was compressed into a dense pellet using a 16-mm-diameter Teflon mold under 440 MPa for 2 minutes, and the resulting pellet was mounted onto a Rigaku in-situ XRD stage (**Fig. R21**) and was subsequently mounted onto a Rigaku in-situ XRD stage for ASSBs (**Fig. R21**), with the cathode composite layer positioned closest to the X-ray beam to optimize signal collection.

Fig. R21. Schematic illustration of *in-situ* XRD module for ASSBs. a) Cross-sectional images and b) schematic diagram for electrochemical operation of *in-situ* XRD module for ASSBs.

Furthermore, reflection-mode XRD (Bragg–Brentano geometry) was employed, which limits the probe depth to the near-surface region—typically ranging from a few to several tens of micrometers—depending on material properties and incidence angle to exclude the signal interference by the solid electrolyte layer and the counter electrode (In/InLi alloy) [M. Birkholz, *Thin Film Analysis by X-ray Scattering*, Wiley-VCH, 2006]. Importantly, aluminum metal (face-centered cubic, JCPDS PDF# 04-0787) exhibits no diffraction peaks near the critical 2θ range of $18\text{--}19^\circ$, which corresponds to the (003) plane reflection of the layered NCM cathode. Its strongest peaks occur at higher angles (e.g., 38.5° , 44.7° , and 65.1° for the (111), (200), and (220) planes, respectively). Likewise, the sulfide-based solid electrolyte LPSCl ($\text{Li}_6\text{PS}_5\text{Cl}$) has characteristic diffraction peaks mainly located at higher angles (e.g., 25° , 29° , 31° , 45°) and does not exhibit any significant reflections in the $18\text{--}19^\circ$ range associated with (003) main peak of NCM [JCPDS PDF# 29-0812]. Thus, no peak overlap or interference from either the Al current collector or the LPSCl electrolyte is expected or observed in the angular region critical for monitoring the structural evolution of the cathode active material for (003) plane.

Accordingly, we confirm that the diffraction signals captured during our *in-situ* measurements arise predominantly from the cathode particles, with negligible contributions from the surrounding components. These experimental considerations ensure that the observed phase evolution and peak shifts can be reliably attributed to the cathode material itself. As response to the reviewer’s comment, we revised manuscript as follow:

Original text (Page 10, line 13)

We investigated the reaction heterogeneity in cathodes with and without surface modification using *in-situ* X-ray diffraction (XRD) and transmission X-ray microscopy (TXM).^{39,40} This analysis, depicted in **Fig. 2**, elucidated the impact of chemical degradation on cathode particle reaction behavior in ASSBs.

Revised text (Page 11, line 6)

We investigated the reaction heterogeneity in cathodes with and without surface modification using *in-situ* X-ray diffraction (XRD) and transmission X-ray microscopy

(TXM).^{39,40} To gain a more detailed understanding of the cathode behavior, we specifically analyzed the (003) reflection of the layered oxide structure within the 2θ range of 18–19°, using *in-situ* XRD in reflection mode (Supplementary Fig. 11). This analysis, depicted in Fig. 2, elucidated the impact of chemical degradation on cathode particle reaction behavior in ASSBs.

Added Figure (Supplementary Fig. 11, Revised Supplementary Information, page 14)

Supplementary Fig. 11. a) Cross-sectional images and b) schematic diagram for electrochemical operation of *in-situ* XRD module for ASSBs.

Comments:

3. Information such as mass loading of active materials and test temperature of the all-solid-state batteries mentioned in this manuscript should be given and labeled in figures for readers' better understanding.

Author reply:

We sincerely thank the reviewer for the helpful and constructive comment. We fully agree that clearly presenting the mass loading of active materials and test temperature is essential for ensuring the clarity and interpretability of electrochemical data in all-solid-state battery (ASSB) research.

In response, we have revised the manuscript to include clear labeling and added the relevant information on active materials mass loading, test temperature, and other experimental details for improved readability.

We really thank the reviewer's comment regarding the clarity of manuscript, and we revised the manuscript as follows:

Original figure (Fig. 1, Original Manuscript, page 34)

Original caption (Fig. 1, Original Manuscript, page 34)

Fig. 1. Chemical degradation mitigation through surface modification and corresponding electrochemical properties. a) Current–time curves of the cathode-SE composite (65:35 wt%) under a polarization of 200 mV in a steel|NCM composite|steel ion-blocking cell, and b) current–time curves of the cathode-SE composite (65:35 wt%) under a polarization of 50 mV in an electron-blocking cell, composed of In/InLi|LPSCI|NCM composite|LPSCI|In/InLi, for both bare NCM and LiDFP NCM. TOF-SIMS 3D depth spectra and 3D images of c) bare NCM and d) LiDFP NCM for S^- , LiS^- , PS_3^- , and NiO_2^- anion after the first cycling. Comparison of e) rate capability of bare NCM and LiDFP NCM at varying C-rates: C/10, C/4, C/2, 1C, 2C, 3C, and back to C/10. f) Long-term cycle performance at C/2 between bare NCM and LiDFP NCM at 25 °C under a stack pressure of 120 MPa.

Revised Figure (Fig. 1, Revised Manuscript, page 38)

Revised caption (Fig. 1, Revised Manuscript, page 38)

Fig. 1. Chemical degradation mitigation through surface modification and corresponding electrochemical properties. a) Current–time curves of the cathode-SE composite (65:35 wt%) under a polarization of 200 mV in a steel|NCM composite|steel ion-blocking cell, and b) current–time curves of the cathode-SE composite (65:35 wt%) under a polarization of 50 mV in an electron-blocking cell, composed of In/InLi|LPSCI|NCM composite|LPSCI|In/InLi, for both bare NCM and LiDFP NCM. TOF-SIMS 3D depth spectra and 3D images of c) bare NCM and d) LiDFP NCM for S^- , LiS^- , PS_3^- , and NiO_2^- anion after the first cycling. Comparison of e) rate capability of bare NCM and LiDFP NCM at varying C-rates: C/10, C/4, C/2, 1C, 2C, 3C, and back to C/10 and subsequent f) long-term cycle performance at C/2 between bare NCM and LiDFP NCM at 25 °C under a stack pressure of 120 MPa with 7.4 mg/cm^2 active material mass loading.

Original figure (Fig. 4, Original Manuscript, page 38)

Original caption (Fig. 4, Original Manuscript, page 38)

Fig. 4. Quantification of microstructural changes due to chemical degradation. Histogram of labeled cathode particles for a) surface area, b) volume, and c) surface area to volume ratio. d) Comparative analysis of porosity and ionic tortuosity between bare NCM and LiDFP NCM within the reconstructed 3D composite electrode. e) Comparison of cathode surface area versus contact loss area and f) comparison of contact loss area versus adjacent pore volume for each labeled cathode particle in both bare NCM and LiDFP NCM. g) GITT and OCV curves of bare NCM and LiDFP NCM during the first discharge cycle and corresponding h) $\left(\frac{\Delta E_s}{\Delta E_t}\right)^2$ profiles for each GITT step of bare NCM and LiDFP NCM. i) Electrochemical cycle performance of bare NCM and LiDFP NCM at C/10 ($1C = 1.36 \text{ mA/cm}^2$) rate and 25°C under varying stack pressure of 2.6, 3.3, and 6.5 MPa.

Revised figure (Fig. 4, Revised Manuscript, page 42)

Revised caption (Fig. 4, Revised Manuscript, page 42)

Fig. 4. Quantification of microstructural changes due to chemical degradation. Histogram of labeled cathode particles for a) surface area, b) volume, and c) surface area to volume ratio. d) Comparative analysis of porosity and ionic tortuosity between bare NCM and LiDFP NCM within the reconstructed 3D composite electrode. e) Comparison of cathode surface area versus contact loss area and f) comparison of contact loss area versus adjacent pore volume for each labeled cathode particle in both bare NCM and LiDFP NCM. g) GITT and OCV curves of bare NCM and LiDFP NCM during the first discharge cycle and corresponding h) $\left(\frac{\Delta E_s}{\Delta E_t}\right)^2$ profiles for each GITT step of bare NCM and LiDFP NCM. i) Electrochemical cycle performance of bare NCM and LiDFP NCM at C/10 (1C = 1.36 mA/cm²) rate and 25 °C under varying stack pressure of 2.6, 3.3, and 6.5 MPa with 7.4 mg/cm² active material mass loading.

Comments:

4. The in/ex-situ characterizations of the composite cathodes of all-solid-state batteries described in this manuscript are impressive. However, considering the air sensitivity of sulfide electrolytes, how does the authors eliminate the above interference during characterizations to ensure the validity of data? Do the current test methods meet the requirements of air tightness? Relevant instructions should be given.

Author reply:

We sincerely thank the reviewer for raising this important and highly relevant concern regarding the air sensitivity of sulfide-based solid electrolytes and the potential impact on the accuracy of ex-situ characterization results. To address this issue, we would like to emphasize that all cell fabrication, handling, disassembly, and sample preparation steps were strictly performed inside an argon-filled glove box (O_2 , $H_2O < 0.1$ ppm). This includes cathode composite mixing, solid electrolyte pressing, full-cell assembly, and post-cycling disassembly for characterization. These procedures ensured that the solid electrolyte and its interfaces were never exposed to ambient air throughout all stages of cell construction and operation. However, regarding sample transfer for ex-situ characterization (such as ToF-SIMS, FIB-SEM, and etc.), we fully acknowledge that brief air exposure (typically <1 minute) can occur during the mounting of samples onto analysis platforms. To evaluate and minimize this potential effect, we conducted a systematic comparison experiment using a standard XPS sample holder, and an air-tight vacuum transfer holder that preserves inert conditions up to the point of analysis as shown in **Fig. R22**.

Fig. R22. S 2p XPS spectra for LPSCl solid electrolyte with and without vacuum transfer module.

In this control experiment, we analyzed Li_6PS_5Cl samples stored under identical conditions and transferred *via* the two different holder systems. As shown in **Fig. R22**, The S 2p XPS spectra for both transfer conditions showed no observable differences in the degree of surface oxidation of sulfide SE which is prone to oxidized in air condition. This confirms that sub-

minute air exposure during sample mounting on analytical tools did not interrupt any precise understanding for valid characterizations. Therefore, we are confident that our sample handling and transfer protocols were sufficiently robust to preserve the chemical state of the sulfide electrolyte during analysis. This rigorous control ensures that the data presented in this manuscript accurately reflect the true electrochemical and interfacial properties of the materials under study.

We really thank the reviewer's comment regarding the reliability of characterization, and we revised the manuscript for clarity as follows:

Original text (Page 23, line 3)

Characterization of ASSBs

The microstructure of the NCM cathode with the LiDFP coating was examined using HR-TEM (JEM-2100F, JEOL). Cross-sectional TEM analysis samples were prepared using FIB (NX2000, Hitachi). The presence of the LiDFP coating layer was confirmed using TOF-SIMS (TOF-SIMS 5, ION TOF GmbH) with a pulsed Bi³⁺ ion beam (25 keV) in high current mode.

Revised text (Page 24, line 16)

Characterization of ASSBs

To minimize exposure to ambient air and ensure the accuracy of analytical results, all sample preparation steps were conducted in an argon-filled glove box with oxygen and moisture levels maintained below 1 ppm. Sample transfer to the analysis instruments was carried out either under inert argon atmosphere or via vacuum-sealed containers to prevent ambient contamination. The microstructure of the NCM cathode with the LiDFP coating was examined using HR-TEM (JEM-2100F, JEOL). Cross-sectional TEM analysis samples were prepared using FIB (NX2000, Hitachi). The presence of the LiDFP coating layer was confirmed using TOF-SIMS (TOF-SIMS 5, ION TOF GmbH) with a pulsed Bi³⁺ ion beam (25 keV) in high current mode.

Comments:

5. During FIB–SEM analysis, the thin slices are removed by ionic etching, considering the different properties of components in composite cathodes, especially the LiDFP coating layer with low Yong's modulus as authors mentioned, does etching rates of them will be different? And does this eventually create excess porosity during the reconstruction of digital twinned 3D model? The detailed verifications and descriptions need to be added.

Author reply:

We greatly appreciate the reviewer's insightful question regarding the fidelity of FIB–SEM-based 3D reconstruction and its potential sensitivity to component-specific etching behavior. This is a technically important consideration, especially when analyzing composite cathode

structures that involve soft surface coatings such as LiDFP, which may differ in mechanical properties from the bulk material.

To address this concern, we carried out the following validation procedures:

1. Baseline Comparison Using Fresh-State Samples (Without Electrochemical Cycling)

As shown in **Fig. R23** (denoted as Supplementary Fig. 2, Supplementary Information), we compared FIB–SEM images of fresh (uncycled) cathode composites: One with no coating (bare NCM), and one with the LiDFP coating material. Despite the presence of the coating layer, no significant difference in apparent porosity or slice contrast was observed in 2D cross-sectional images as informed by mean pore area and distribution of pore size shown in **Fig. R23c**. This suggests that the etching rates of the constituent phases—including the thin LiDFP layer—did not lead to artificial pore formation or preferential material removal during milling.

Fig. R23. Cross-sectional SEM images of a) bare NCM composite and b) LiDFP NCM composite right after the cell fabrication and corresponding c) mean pore area ratio and distribution of pore for fresh cells. Red dots represent the pore at the cathode composite.

2. Consideration of Voxel Size vs. Coating Thickness

As also noted by the reviewer, the LiDFP layer is extremely thin, approximately 5–10 nm, as confirmed via high-resolution TEM in the Supplementary Fig. 2 (see **Fig. R24**). In contrast, the voxel resolution in our digital twin 3D reconstruction is 46.5 nm (**Fig. R25**), which is approximately 4–9 times larger than the coating thickness. Therefore, even in the event of a slightly faster or slower etching rate of the LiDFP coating, the difference falls well below the spatial resolution limit of our 3D imaging model. As such, this discrepancy would not be captured as a distinct pore or void in the digital twin reconstruction, and would not contribute to excess porosity artifacts in the final 3D analysis.

Fig. R24. a) Cross-sectioned STEM image of NCM surface coated with 0.3 wt% LiDFP, denoted as LiDFP NCM. b) Cross-sectioned STEM image and corresponding EDS mapping for Ni and P atoms. TOF-SIMS surface spectra of LiDFP additive, bare NCM, and LiDFP NCM of c) PO_2^- anion and d) PO_2F_2^- anion.

Fig. R25. Cross-sectional SEM images used for U-net method with pixel size of 46.52 nm.

In summary, we believe that the impact of differential etching behavior—particularly due to the thin LiDFP layer—is negligible within the resolution of our FIB–SEM setup. The validation through fresh-state imaging and voxel-to-coating thickness ratio reinforces the reliability of our digital twinned reconstruction.

We really thank the reviewer’s comment regarding the clarity of manuscript, and we revised the manuscript as follows:

Original text (Page 23, line 18)

For the FIB 3D images, ASSB pellets retrieved from a 13-mm-diameter Teflon body and pressed under a uniaxial stack pressure of approximately 13 MPa were subjected to FIB–SEM imaging (PP3010, Quorum). These pellets were positioned at a 36° tilt to ensure that the cathode composite faced the ion beam perpendicularly. Initially, the ion beam created islands around the target area, and an SEM image was taken to capture the 2D morphology

of the cathode composite. Subsequently, thin slices of 97.52 nm, with a pixel size of 46.52 nm, were sequentially removed from the cross-section using the ion beam. Each removal was followed by SEM image capture, repeated multiple times.

Revised text (Page 25, line 12)

For the FIB 3D images, ASSB pellets retrieved from a 13-mm-diameter Teflon body and pressed under a uniaxial stack pressure of approximately 13 MPa were subjected to FIB-SEM imaging (PP3010, Quorum). These pellets were positioned at a 36° tilt to ensure that the cathode composite faced the ion beam perpendicularly. Initially, the ion beam created islands around the target area, and an SEM image was taken to capture the 2D morphology of the cathode composite. Subsequently, thin slices of 97.52 nm, with a pixel size of 46.52 nm, were sequentially removed from the cross-section using the ion beam. Each removal was followed by SEM image capture, repeated multiple times. To ensure the fidelity of 3D digital twin reconstruction, we evaluated potential artifacts arising from differential ion etching of component phases—particularly the mechanically soft LiDFP coating layer. Comparative FIB-SEM imaging of fresh-state cathode composites (as shown in Supplementary Fig. S3) revealed no significant differences in pore contrast or morphology. Additionally, the voxel size of the reconstruction (46.5 nm) substantially exceeds the LiDFP coating thickness (5–10 nm), minimizing the likelihood of artificial porosity resulting from coating-specific etching. These validations confirm that the presence of LiDFP does not compromise the accuracy of porosity or connectivity analysis in the reconstructed model.